# IGSEEK: FAST AND ACCURATE ANTIBODY DESIGN VIA STRUCTURE RETRIEVAL

## ABSTRACT

Recent advancements in protein design have leveraged diffusion models to generate structural scaffolds, followed by a process known as protein inverse folding, which involves sequence inference on these scaffolds. However, these methodologies face significant challenges when applied to hyper-variable structures such as antibody Complementarity-Determining Regions (CDRs), where sequence inference frequently results in non-functional sequences due to hallucinations. Distinguished from prevailing protein inverse folding approaches, this paper introduces IgSeek, a novel structure-retrieval framework that infers CDR sequences by retrieving similar structures from a natural antibody database. Specifically, IgSeek employs a simple yet effective multi-channel equivariant graph neural network to generate high-quality geometric representations of CDR backbone structures. Subsequently, it aligns sequences of structurally similar CDRs and utilizes structurally conserved sequence motifs to enhance inference accuracy. Our experiments demonstrate that IgSeek not only proves to be highly efficient in structural retrieval but also outperforms state-of-the-art approaches in sequence recovery for both antibodies and T-Cell Receptors, offering a new retrieval-based perspective for therapeutic protein design.

## 1 INTRODUCTION

Antibodies, known for their high specificity and affinity, have emerged as pivotal therapeutic agents in the treatment of complex diseases, including cancer (Adams & Weiner, 2005), autoimmune disorders (Feldmann & Maini, 2003), and infectious diseases (Abraham, 2020). In 2023, the global best-selling drug was Keytruda, a cancer treatment antibody, with sales reaching $25 billion, surpassing Humira, another antibody used for treating rheumatoid arthritis, which had dominated the market for the past decade (Dunleavy, 2024). Traditionally, the discovery of antibodies has predominantly relied on immunizing animals with antigens (Van Wauwe et al., 1980) or employing various display techniques such as phage (MacCallum et al., 1996) and yeast displays (Chao et al., 2006). However, these approaches face significant challenges when dealing with structurally intricate proteins, which are difficult to express in a soluble and functional form. Additionally, even when numerous candidate antibodies are generated through these techniques, they may not necessarily bind to the desired domain or exhibit therapeutic efficacy.

To overcome these limitations, deep learning models have been introduced to design synthetic antibodies by learning from natural antibody-antigen complexes (Luo et al., 2022; Jin et al., 2022; Kong et al., 2023b;a; Bennett et al., 2024). Despite significant strides in protein design (Dauparas et al., 2022; Hsu et al., 2022; Notin et al., 2024), antibodies present a distinct challenge for deep learning due to the high flexibility of their binding regions, known as complementarity-determining regions (CDRs). Encouraged by the success of RFdiffusion (Watson et al., 2023) in monomeric protein and binder designs, Bennett et al. (2024) fine-tuned the RFdiffusion model using structural data of antibody-antigen complexes to design antibodies targeting predetermined antigen epitopes. Although functional assays and structural determination experiments confirmed that this method could produce antibodies binding to predetermined epitopes, the success rate was extremely low.

One reason for the low success rate of AI-designed antibodies is the occurrence of hallucinations during sequence inference given the backbone structure, also known as the protein inverse folding problem (Dauparas et al., 2022; Hsu et al., 2022; Gruver et al., 2023; Gao et al., 2023b). The amino

Table 1: Settings of Different Antibody (Ab) Design Tasks.

| Category | Input | | | Output | |
|---|---|---|---|---|---|
| | Ab Framework | Ab CDR | Antigen | CDR Structure | CDR Sequence |
| Antibody Co-design | ✓ | ✓ | ✓ | ✓ | ✓ |
| Antibody Inverse Folding | ✓ | ✓ | ✗ | ✗ | ✓ |
| Antibody Sequence Design | ✗ | ✓ | ✗ | ✗ | ✓ |

acid sequences inferred through methods like ProteinMPNN (Dauparas et al., 2022) and ESM-IF1 (Hsu et al., 2022) may not fold into the desired structures in real biological systems. More critically, there are currently no effective computational methods to reduce these hallucinations, aside from conducting time-consuming, labor-intensive, and expensive wet-lab experiments for validation. Typically, using independent structure prediction models to fold and verify the inferred sequences cannot effectively eliminate non-functional sequences caused by hallucinations. That is because even state-of-the-art models exhibit structural deviations of 1 to 3 Å and have low confidence in predicting the structures of antibody CDRs.

To deal with the challenge of hallucinations arising from previous models, we propose an antibody CDR sequence design framework from a novel perspective of similar structure retrieval. Our framework, named as IgSeek (Ig for Immunoglobulin, a.k.a. antibody), is enlightened by a noteworthy empirical discovery made 25 years ago, which revealed that antibodies exhibit a limited set of canonical structures within 5 out of 6 CDRs despite the vast diversity in sequences, and that certain CDR conformations are scaffolded by a few highly conserved residues (Chothia et al., 1989). Further inspired by retrieval-augmented prediction for hallucination reduction in protein structure prediction (Jumper et al., 2021), and natural language generation (Gao et al., 2023a), IgSeek leverages neural retrieval in an antibody database to retrieve structurally similar sequence templates of CDR, and ensembles the queried templates for sequence prediction. In summary, the contributions of this paper are as follows:

- We propose a novel framework, IgSeek, that utilizes antibody structure retrieval to enhance the accuracy and reliability of AI-driven antibody design.

- IgSeek employs a Multi-channel Equivariant Graph Neural Network to construct an antibody structure database for isomorphic structure retrieval, where the structural representation is invariant to E(3) transformations.

- Our extensive experiments demonstrate that IgSeek substantially outperforms state-of-the-art competitors by a large margin regarding sequence recovery rate and inference speed.

## 2 RELATED WORK

**Protein Structure Retrieval.** With the growth of the volume of protein structures, structure retrieval has become a critical task in protein data management. AlphaFind (Procházka et al., 2024) is a web tool designed to identify structurally similar proteins in AlphaFold Database (Varadi et al., 2022) by compressing data from ∼23 TB to ∼20 GB using vector embeddings, narrowing down candidates with a neural network. The similarity of the search result is evaluated by US-align (Zhang et al., 2022). Another state-of-the-art method, FoldSeek (Van Kempen et al., 2024), accelerates protein structure searches by representing tertiary amino acid interactions as sequences over a 3D interaction structural alphabet, which derives from vector quantization by VQ-VAE (van den Oord et al., 2017). However, the representation only models the structure of two contiguous residues in a chain.

**Protein Inverse Folding.** Protein inverse folding aims to predict diverse sequences that can fold into a given protein structure. ProteinMPNN (Dauparas et al., 2022) is a deep learning–based method for protein sequence design that excels in both in silico and experimental evaluations. By leveraging a message-passing neural network with enhanced input features and edge updates, ProteinMPNN is capable of designing monomers, cyclic oligomers, protein nanoparticles, and protein-protein interfaces, rescuing previously failed designs generated by Rosetta (Adolf-Bryfogle et al., 2018; Baek et al., 2021) or AlphaFold (Jumper et al., 2021). ESM-IF1 (Hsu et al., 2022) employs a sequence-to-sequence Transformer to predict protein sequences from backbone atom coordinates.

**Antibody Inverse Folding.** AbMPNN (Dreyer et al., 2023) inherits the model architecture of ProteinMPNN, and trains an antibody-specific variant for antibody design. It outperforms generic protein design models in sequence recovery and structure robustness, especially for hyper-variable CDR-H3 loops. AntiFold (Høie et al., 2024) is an antibody-specific inverse folding model, which is fine-tuned on ESM-IF1, with both solved and predicted antibody structures. However, it should be emphasized that Antifold infers CDR sequences based on the structure of the variable domain and the sequence of the framework regions. Consequently, the accuracy of CDR sequence inference is influenced not only by the structure of the CDRs but also by the sequence and structural information of the framework regions. Previous studies utilizing antibody sequence language models without structural information have demonstrated that the sequence of the framework regions can partially predict the CDR sequences, particularly for relatively conserved residues. As a result, the requirement for the framework sequence as input complicates the inference of CDR sequences that can bind to different antigens while maintaining an identical framework.

**Antibody Co-Design.** In recent years, deep learning models have emerged as powerful data-driven approaches for antibody design. RefineGNN (Jin et al., 2022) is the first structure sequence co-design method that alternatively predicts the atom coordinates and residue types in CDRs by auto-regression. DiffAb (Luo et al., 2022) and IgGM (Wang et al., 2024) utilize diffusion models to generate the structure and sequence of CDRs based on the framework regions and the target antigen, with DiffAb oriented for specific antigens. MEAN (Kong et al., 2023b) and dyMEAN (Kong et al., 2023a) employ graph neural networks to predict the structure and sequence of CDRs. Table 1 presents a comparative analysis of various antibody design task configurations.

## 3 PRELIMINARIES AND PROBLEM FORMULATION

Antibodies are special Y-shape proteins, whose binding specificity is characterized by CDRs in the variable regions (refer to Appendix A). We represent the 3D structure of a CDR as a geometric graph $G = (V, E)$ with node set $V$ and edge set $E$ (Jing et al., 2021; Jin et al., 2022; Zhang et al., 2023). Each node $v_i \in V$ denotes an amino acid residue, associated with a multi-channel 3D coordinate matrix $\boldsymbol{X}_i \in \mathbb{R}^{c \times 3}$, where $c$ is the channel size, i.e., the number of atoms in the residue $v_i$. In this paper, we consider the four backbone atoms $\{N, C_\alpha, C, O\}$ that are independent to residue type, i.e., $c = 4$. Each edge $e_{ij} \in E$ denotes an interaction between $v_i$ and $v_j$, if the Euclidean distance between their $C_\alpha$ atoms is within a threshold $\theta$. The neighborhood of a node $v_i$, denoted as $\mathcal{N}_i$, consists of the adjacency nodes of $v_i$, that is, $\{v_j | (v_i, v_j) \in E\}$.

**CDR sequence design.** Given the structure $G = (E, V)$ of a CDR and the multi-channel 3D coordinate of each residue, in this paper, we aim to reconstruct the corresponding sequence of the CDR, denoted as $\boldsymbol{s} = \{s(i) | i \in [1, \cdots, |V|]\}$, where $s(i)$ is the amino acid type of residue $v_i$.

E(3) Equivalence is an important property in modeling the 3D structures (Fuchs et al., 2020; Batzner et al., 2022; Liao & Smidt, 2023). Formally, let $\mathcal{X}$ and $\mathcal{Y}$ be two vector spaces, with $T_{\mathcal{X}}(g) : \mathcal{X} \to \mathcal{X}$ and $T_{\mathcal{Y}}(g) : \mathcal{Y} \to \mathcal{Y}$ representing two sets of transformations for the abstract group $g \in E(3)$. A function $\phi : \mathcal{X} \to \mathcal{Y}$ is E(3) Equivariant to $g$ if it satisfies the following condition:

$$\phi(\{T_{\mathcal{X}}(g)\boldsymbol{x}_i, \boldsymbol{h}_i\}_{i=1}^n) = T_{\mathcal{Y}}(g)\phi(\{\boldsymbol{x}_i, \boldsymbol{h}_i\}_{i=1}^n), \tag{1}$$

where $\boldsymbol{x}_i \in \mathbb{R}^3$ denotes the input 3D coordinates and $\boldsymbol{h}_i \in \mathbb{R}^d$ is the $d$-dimensional features of a node, respectively. This inductive bias guarantees that $\phi$ preserves equivariant transformation regarding transformation of the coordinate system in E(3) group (Satorras et al., 2021; Huang et al., 2022; Liao & Smidt, 2023). A typical example for this transformation operation in the space $\mathcal{X}$ is given by $T_{\mathcal{X}}(g)\boldsymbol{x}_i^{(0)} = \boldsymbol{R}\boldsymbol{x}_i^{(0)} + \boldsymbol{b}$, where $\boldsymbol{R} \in \mathbb{R}^{3 \times 3}$ is an orthogonal matrix and $\boldsymbol{b}$ is the bias term.

To achieve equivalence, equivariant graph neural networks are proposed (Satorras et al., 2021; Huang et al., 2022; Kong et al., 2023a;b), which follows a general message-passing framework as shown in Eq. 2-4. Here, $\boldsymbol{m}_{j \to i}^{(l)}$ denotes the messages propagated from node $v_j$ to $v_i$, and $d_{ij}^{(l-1)} = dist(v_i, v_j)$ denotes the Euclidean distance between $v_i$ and $v_j$, and $\boldsymbol{x}_{ij}^{(l-1)}$ denotes co-

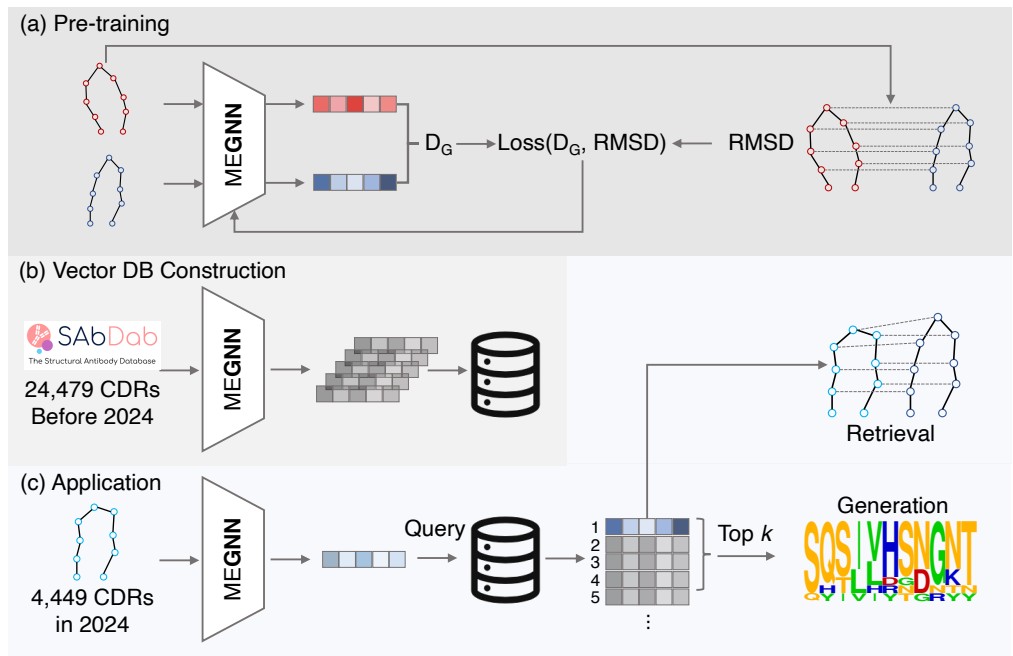

Figure 1: The Framework of IgSeek: (a) Pre-train an MEGNN encoder by a self-supervised learning task. (b) Construct a CDR vector database. (c) Sequence generation by K-NN search.

ordinate differences between $v_i$ and $v_j$ at the $(l-1)$-th layer.

$$m_{j \to i}^{(l)} = \psi_1 \left( h_i^{(l-1)}, h_j^{(l-1)}, x_{ij}^{(l-1)}, d_{ij}^{(l-1)} \right), \tag{2}$$

$$h_i^{(l)} = \psi_2 \left( h_i^{(l-1)}, \sum_{v_j \in \mathcal{N}_i} m_{j \to i}^{(l)} \right), \tag{3}$$

$$x_i^{(l)} = \psi_3 \left( x_i^{(l-1)}, x_{ij}^{(l-1)} \sum_j \psi_4 \left( \sum_{v_j \in \mathcal{N}_i} m_{j \to i}^{(l)} \right) \right). \tag{4}$$

The functions $\{\psi_1, \psi_2, \psi_3, \psi_4\}$ are equivariant transformations, typically implemented as Multi-Layer Perceptrons (MLPs) to leverage the universal approximation (Funahashi, 1989; Cybenko, 1989; Hornik, 1991). In this process, the feature $h_i^{(l)}$ remains E(3) invariant, while the coordinate $x_i^{(l)}$ is E(3) equivariant.

## 4 IGSEEK: OUR METHODOLOGY

In this section, we present our retrieval-based CDR sequence design framework, IgSeek. The gist of IgSeek for structure-to-sequence generation is isomorphic structure retrieval, which allows for the exploration of a large and diverse antibody CDR structure database. Fig. 1 illustrates the framework of IgSeek. Given an antibody CDR database where both structures and sequences are available, IgSeek first constructs a CDR vector database, where vector embeddings index the structural proximity of the CDRs. In this offline stage, we pre-train a *Multi-channel Equivariant Graph Neural Network (MEGNN)* to encode the structure of CDR loops into fixed-length vectors within the CDR database. Specifically, MEGNN aligns the spatial structure distance between pairs of CDRs with equal lengths and similar conformations. Subsequently, for a CDR structure $G$ whose sequence is to be predicted, we first deploy the pre-trained MEGNN to generate an embedding $h_G$ for $G$. $h_G$ then serves as the search key to query the *$K$-nearest neighbors ($K$-NN)* structurally similar CDR loops in the vector database. Finally, the $K$-NN results, associated with their corresponding residue sequences, are collected for predicting the sequence of $G$ by ensemble and Bernoulli sampling. In

the following, we will present the model design of the MEGNN encoder in Section 4.1, discuss the learning objective and the sequence prediction in Section 4.2, followed by model analysis in Section 4.3.

## 4.1 MULTI CHANNEL EQUIVARIANT ENCODER

Recall that each amino acid residue $v_i$ is represented by its four backbone atoms, thereby we extend the general single-channel EGNN layer (Satorras et al., 2021; Huang et al., 2022) to a multi-channel layer, with each channel corresponding to a specific atom. Unlike existing approaches (Kong et al., 2023a; Høie et al., 2024) that leverage domain knowledge of the well-conserved antibody backbone structure, our MEGNN encoder generates CDR embeddings exclusively based on the antibody CDR structure, without relying on any prior backbone knowledge.

For a 3D CDR structure $G$, the MEGNN encoder takes the initial features of each residue $v_i$, denoted as $\boldsymbol{h}_i^{(0)} \in \mathbb{R}^d$, along with the perturbed coordinates $\hat{\boldsymbol{X}}_i \in \mathbb{R}^{c \times 3}$ as input. Here, $c$ denotes the number of atoms, which is set to $c = 4$, and $\boldsymbol{h}_i^{(0)}$ is initialized by a uniform distribution. $\hat{\boldsymbol{X}}_i = \boldsymbol{X}_i + \mathcal{N}(0, \sigma)$, where $\mathcal{N}(0, \sigma)$ denotes a small Gaussian noise. This perturbation introduces variability that enhances the robustness of the model.

**Multi-channel Equivariant Message Passing.** The $l$-th layer of MEGNN updates both the node features $\boldsymbol{h}_i^{(l)}$ and coordinates $\boldsymbol{X}_i^{(l)}$ by Eq. 5-8, where $\rho$ is a distance computation function, $\phi_e$, $\phi_X$ and $\phi_h$ are neural network transformations. The update process is defined as follows:

$$\boldsymbol{X}_{ij}^{(l-1)}, \boldsymbol{z}_{ij}^{(l-1)} = \rho\left(\boldsymbol{X}_i^{(l-1)}, \boldsymbol{X}_j^{(l-1)}, e_{ij}\right), \tag{5}$$

$$\boldsymbol{h}_{e_{ij}}^{(l)} = \phi_e\left(\text{CONCAT}\left(\boldsymbol{h}_i^{(l-1)}, \boldsymbol{h}_j^{(l-1)}, \boldsymbol{z}_{ij}^{(l-1)}\right)\right), \tag{6}$$

$$\boldsymbol{X}_i^{(l)} = \phi_X\left(\boldsymbol{X}_i^{(l-1)}, \{\boldsymbol{h}_{e_{ij}}^{(l)}, \boldsymbol{X}_{ij}^{(l-1)} | v_j \in \mathcal{N}_i\}\right), \tag{7}$$

$$\boldsymbol{h}_i^{(l)} = \phi_h\left(\boldsymbol{h}_i^{(l-1)}, \{\boldsymbol{h}_{e_{ij}}^{(l)} | v_j \in \mathcal{N}_i\}\right). \tag{8}$$

Specifically, MEGNN first computes the coordinate differences $\boldsymbol{X}_{ij}^{(l-1)}$ and the square distance $\boldsymbol{z}_{ij}^{(l-1)}$ between each pair of backbone atoms among different residues in $\rho$ (Eq. 5) as below:

$$\boldsymbol{X}_{ij}^{(l-1)} = \boldsymbol{X}_i^{(l-1)} - \boldsymbol{X}_j^{(l-1)}, \quad \boldsymbol{z}_{ij}^{(l-1)} = (\boldsymbol{X}_{ij}^{(l-1)})^\top \boldsymbol{X}_{ij}^{(l-1)}.$$

Subsequently, an edge module $\phi_e$ generates the edge feature $\boldsymbol{h}_{e_{ij}}^{(l)}$ for each edge $e_{ij} = (v_i, v_j) \in E$. In Eq. 6, the node features of $v_i$ and $v_j$, i.e., $\boldsymbol{h}_i^{(l-1)}$, $\boldsymbol{h}_j^{(l-1)}$, along with the fattened coordinate difference ($\boldsymbol{z}_{ij}^{(l-1)}$), are concatenated and transformed by an MLP, generating the output edge feature for the $l$-th layer. Next, the coordinate module $\phi_X$ updates the node coordinates $\boldsymbol{X}_i^{(l)}$ using the updated edge feature $\boldsymbol{h}_{e_{ij}}^{(l)}$ and the coordinate differences $\boldsymbol{X}_{ij}^{(l-1)}$ in Eq. 7. Specifically, for each node $v_i$, $\phi_X$ first computes the message $\boldsymbol{m}_{j \to i}$ propagated from its neighbor $v_j$, and then updates the coordinates $\boldsymbol{X}_i^{(l)}$ of $v_i$ by aggregating the messages from its neighborhood:

$$\boldsymbol{m}_{j \to i} = \text{MLP}\left(\boldsymbol{h}_{e_{ij}}^{(l)}\right) \cdot \boldsymbol{X}_{ij}^{(l-1)}, \quad \boldsymbol{X}_i^{(l)} = \boldsymbol{X}_i^{(l-1)} + \frac{1}{|\mathcal{N}_i|} \sum_{v_j \in \mathcal{N}_i} \boldsymbol{m}_{j \to i}.$$

Finally, the node module $\phi_h$ updates the node representation $\boldsymbol{h}_i^{(l)}$ by Eq. 8. For each node $v_i$, $\phi_h$ aggregates the features of the adjacent edges into $\boldsymbol{h}_{agg_i}^{(l)}$ and combines the node representation $\boldsymbol{h}_i^{(l-1)}$ from the $(l-1)$-th layer with the aggregated feature using a residual connection (He et al., 2016):

$$\boldsymbol{h}_{agg_i}^{(l)} = \sum_{j \in \mathcal{N}_i} \boldsymbol{h}_{e_{ij}}^{(l)}, \quad \boldsymbol{h}_i^{(l)} = \boldsymbol{h}_i^{(l-1)} + \text{MLP}\left(\text{CONCAT}(\boldsymbol{h}_i^{(l-1)}, \boldsymbol{h}_{agg_i}^{(l)})\right).$$

**CDR Embedding Generation.** After the equivariant message passing through an $L$-layer MEGNN, we employ a READOUT function to aggregate the final node features to generate the representation

of CDR $G$ that consists of $n$ nodes (amino acids) as Eq. 9,

$$h_G = \text{READOUT}(\{h_i^{(L)}\}_{i=1}^n). \tag{9}$$

The READOUT function can be a permutation invariant function, e.g. summation and element-wise mean pooling functions. In our implementation, we set the READOUT function as element-wise mean pooling by default.

## 4.2 LEARNING OBJECTIVE AND SEQUENCE GENERATION

We train the MEGNN encoder by a self-supervised distance prediction task that explicitly aligns pairs of similar CDR in a given database. The goal is to align the structural representation of similar CDR pairs. For a CDR database $\mathcal{B} = \{G_1, G_2, \cdots, G_n\}$, we construct a training dataset $\mathcal{T} = \{(G_i, G_j), \cdots\}$ containing pairs of fixed-length CDRs whose TM-Score, calculated by TM-align (Zhang & Skolnick, 2005), exceeds a specified threshold. Given a pair of CDRs $(G_i, G_j)$, we first generate their representations using MEGNN, denoted as $h_{G_i}$ and $h_{G_j}$, respectively. Next, we predict the *Root Mean Square Deviation (RMSD)* of the two CDR structures by feeding the concatenation of $h_{G_i}$ and $h_{G_j}$ into an MLP decoder as Eq. 10:

$$\widehat{d}(G_i, G_j) = \text{MLP}\left(\text{CONCAT}\left(h_{G_i}, h_{G_j}\right)\right). \tag{10}$$

**Loss Function.** The learning objective is to minimize the Mean Square Error between the predicted distance $\widehat{d}(G_i, G_j)$ and the actual distance $d(G_i, G_j)$ in the training dataset $\mathcal{T}$:

$$\mathcal{L} = \frac{1}{|\mathcal{T}|} \sum_{(G_i, G_j) \in \mathcal{T}} \|\widehat{d}(G_i, G_j) - d(G_i, G_j)\|^2. \tag{11}$$

Here, the actual distance $d(G_i, G_j)$ is computed as the RMSD of the two CDRs for their backbone atoms. Since we do not have prior knowledge of the CDR cluster labels, our approach can be interpreted as an unsupervised geometric learning model. By minimizing the loss function defined in Eq. 11, the model effectively generates CDR embeddings that reflect the structural relationships among the CDRs in the dataset.

**CDR Sequence Generation.** Once the model training is complete, we establish a CDR vector database $\mathcal{Z}$, where each $\text{CDR}_i$ is represented by a triplet $(s_i, G_i, h_{G_i})$ consisting of amino acid sequence $s_i$, its backbone structure graph $G_i$ and its embedding $h_{G_i}$ generated by the MEGNN encoder via Eq. 9. IgSeek is then able to infer the amino acid sequence of a CDR by querying its backbone structure in the database $\mathcal{Z}$. Let $s_q$ denote the query CDR sequence with a length of $L$. At each position $l \in \{1, \cdots, L\}$, the residue $s_q(l)$ is selected from one of the 20 amino acids, denoted as $a_i$ for $i \in \{1, \cdots, 20\}$. Then, the inference of the CDR sequence $s_q$ given its backbone structure $G_q$ follows four steps: *(i)* first, the MEGNN encoder generates the embedding of $G_q$, denoted as $h_{G_q}$. *(ii)* Second, the embedding $h_{G_q}$ is used as the search key to perform a $K$-NN search in the database $\mathcal{Z}$, obtaining a set of $K$ CDRs of equal length $L$, denoted as $\mathcal{Z}_q = \{(s_1, G_1, h_{G_1}), (s_2, G_2, h_{G_2}), \cdots, (s_K, G_K, h_{G_K})\}$. *(iii)* Given the $K$ sequences $\mathcal{S}_q = \{s_1, \cdots, s_K\}$, we derive the probability of amino acid $a_i$ occurring at position $l$ of the predicted sequence $\hat{s}_q$ as follows:

$$p\left(\hat{s}_q(l) = a_i | S_q\right) = \frac{1}{K} \sum_{s_k \in \mathcal{S}_q} \mathbb{I}(s_k(l), a_i),$$

where $\mathbb{I}(s_k(l), a_i) \in \{0, 1\}$ is a binary indicator that equals 1 if the amino acid $a_i$ occurs at the position $l$ of sequence $s_k$, and 0 otherwise. *(iv)* To derive the final inferred sequence $\hat{s}_q$, we sample the amino acid at each position $l$ according to the generated probability distribution:

$$\hat{s}_q(l) \sim p\left(\hat{s}_q(l) | \mathcal{S}_q\right).$$

## 4.3 ANALYSIS

**Model Complexity.** Given a 3D CDR structure represented by $G = (V, E)$, the initialized coordinates, node features, and the graph structure contribute a space complexity of $O(|V| \cdot d + |V| \cdot c +$

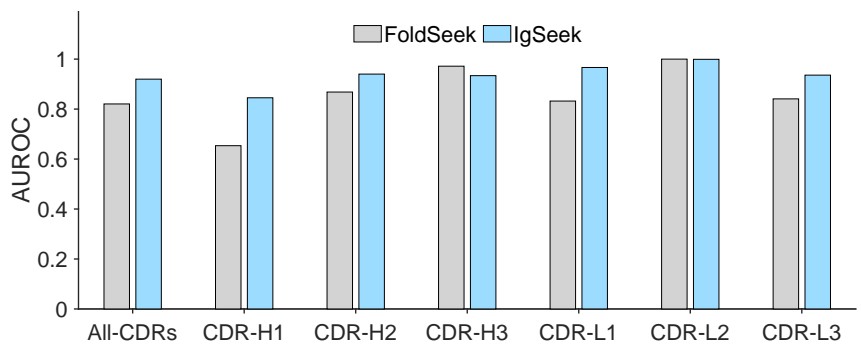

Figure 2: Comparison of IgSeek vs. FoldSeek in CDR Retrieval.

$|E|) = O(|V| \cdot d + |E|)$, where $d$ denotes the hidden dimension of features and $c$ denotes the channel size. In MEGNN, the space complexity is dominated by the edge features, which have a complexity of $O(|E| \cdot d)$, and square distance $\boldsymbol{z}$ with a complexity of $O(|E| \cdot c^2)$. Consequently, the overall space complexity is $O(|E| \cdot d)$, which is linear to the input graph size. Regarding the computational complexity of MEGNN, the dominant component is the edge module $\phi_e$ introduced in Eq. 6, which has a time complexity of $O(|E| \cdot (2d + 3c)^2 + |E| \cdot d^2 + 3c) = O(|E| \cdot d^2)$.

**Coordinate Equivariance and Representation Invariance.** The following theorem shows that MEGNN is E(3) equivariant with respect to the initial coordinate $\boldsymbol{X}_i^{(0)}$ and E(3) invariant with respect to the representations $\boldsymbol{h}$ of the input CDR, respectively.

**Theorem 1.** *For any transformation* $g \in E(3)$, *we have* $\boldsymbol{h}_i, T_{\mathcal{Y}}(g)\boldsymbol{X}_i^{(L)} = MEGNN\left(\boldsymbol{h}_i^{(0)}, T_{\mathcal{X}}(g)\boldsymbol{X}_i^{(0)}, G\right)$, *where* $T_{\mathcal{X}}$ *and* $T_{\mathcal{Y}} := \boldsymbol{R}\boldsymbol{X} + \boldsymbol{b}$ *denotes the transformation of* $\boldsymbol{X}$ *in the input space* $\mathcal{X}$ *(resp. output space* $\mathcal{Y}$*),* $\boldsymbol{R}$ *is an orthogonal matrix, and* $\boldsymbol{b}$ *is the bias.*

The theorem indicates that MEGNN can be generalized to arbitrary E(3) group operations (refer to Section 3), which showcases the data efficiency of MEGNN. The formal proof of Theorem 1 is provided in Appendix C.

## 5 EXPERIMENTAL STUDIES

In this section, we give the test setting in Section 5.1 and report our comprehensive experiments in the following facets: (1) Compare IgSeek with structure retrieval approach (Section 5.2) (2) Compare IgSeek with sequence design approaches (Section 5.3) (3) Investigate the CDR structure representations encoded by MEGNN by visualization (Section 5.4) (4) Study the sequence generation by a case study (Section 5.5).

### 5.1 EXPERIMENTAL SETUP

**Datasets.** We evaluate our IgSeek and other baselines using both solved and predicted antibody structures. The training set consists of CDR pairs sampled from $11,023$ solved CDR loops in the *Structural Antibody Database (SAbDab)* (Dunbar et al., 2013; Schneider et al., 2021). To construct the CDR vector database, we utilize $24,479$ solved CDR loops from SAbDab before January 1, 2024 (SAbDab-before-2024). In addition, $4,449$ solved CDR loops released between January 3, 2024 and May 29, 2024 from SAbDab (SAbDab-2024) serve as the test set to evaluate the performance of IgSeek and its competitors. In addition to the solved antibody structures from SAbDab, we also conduct experiments on $5,111$ CDR loops from the *Structural T-Cell Receptor Database (STCRDab)* (Leem et al., 2018) to evaluate the model generalization ability. Furthermore, we evaluate the model efficiency using $5,000$ predicted CDR-H3 loops from the *Observed Antibody Space (OAS-H3)* (Kovaltsuk et al., 2018; Olsen et al., 2022). More details of each dataset can be found in Appendix B.

**Competitors.** We compare our IgSeek against 5 state-of-the-art models. More details can be found in Appendix F.

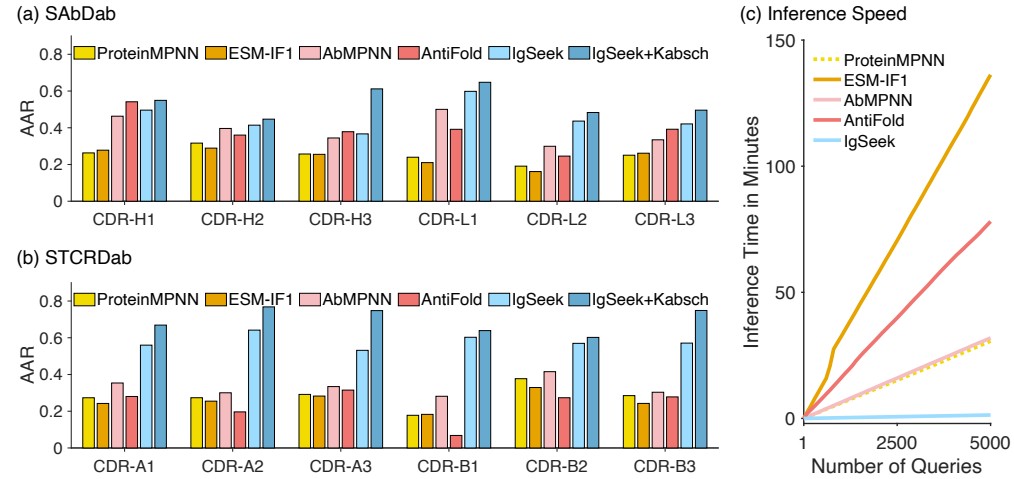

Figure 3: The Comparison of Average AAR and Inference Speed. (a) AAR in SAbDab-2024 Dataset. (b) AAR in STCRDab. (c) Inference Speed

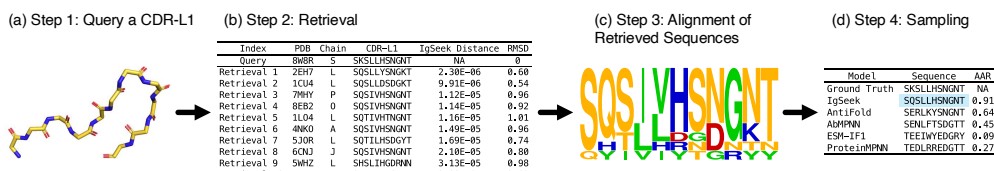

Figure 4: Case study using 8W8R CDR-L1 as an example.

- Structure retrieval model: FoldSeek (Van Kempen et al., 2024).
- Protein and antibody sequence design model: ProteinMPNN (Dauparas et al., 2022), ESM-IF1 (Hsu et al., 2022), AbMPNN (Dreyer et al., 2023), and AntiFold (Høie et al., 2024).

**Parameters.** To ensure a fair comparison, we obtain the source code of all competitors from GitHub and use the default parameter settings suggested by their authors. The implementation details of IgSeek can be found in Appendix D. Additionally, we incorporate a variant of IgSeek that uses RMSD as a secondary sorting metric, denoted as IgSeek+Kabsch.

## 5.2 CDR STRUCTURE RETRIEVAL

In this set of experiments, we compare IgSeek with the state-of-the-art structure searching model, FoldSeek, by examining the quality of the retrieved isomorphic structures. Specifically, for a given query CDR $q$, the retrieved CDR $r$ is considered a positive instance if their RMSD is less than 1 Å. To ensure the robustness of our evaluation, we omit any query CDR for which there are no candidates in the CDR database with a distance of less than 1 Åfrom the query. This strategy allows us to focus on instances where meaningful comparisons can be made, thereby enhancing the result reliability.

Fig. 2 presents the experimental results of IgSeek and FoldSeek, illustrating the model performance on the retrieved sequences using the AUROC metric. As we can observe, IgSeek outperforms Fold-Seek on four types of CDR loops while maintaining comparable performance on CDR-H3 and CDR-L1, indicating its capability of identifying structurally similar CDRs across diverse CDR loops. It is worth noting that IgSeek achieves a 2.6x speed-up in structure retrieval time compared to Fold-Seek. Since this improvement in speed does not come at the cost of accuracy, it demonstrates that IgSeek strikes a superior trade-off between efficiency and accuracy. The ability to quickly retrieve high-quality structural matches can greatly enhance workflows in antibody design, as we will show in Section 5.3.

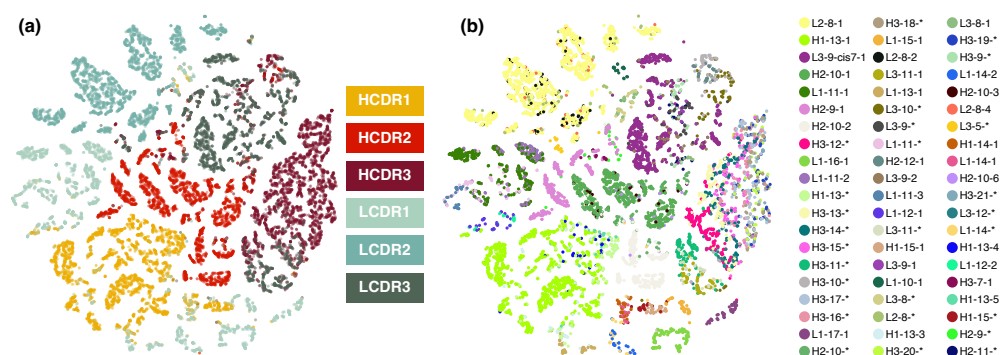

Figure 5: Embeddings of CDRs in the SAbDab-before-2024 datasets projected onto 2D Space.

## 5.3 CDR SEQUENCE DESIGN

**Sequence Recovery.** In the first set of experiments, the MEGNN in IgSeek is trained on the the SAbDab-before-2024 dataset to construct the CDR vector database. Subsequently, the trained MEGNN is utilized to generate embeddings for the CDRs in the SAbDab-2024 dataset. For each query CDR in the SAbDab-2024 dataset, we retrieve the top-10 nearest neighbors from the SAbDab-before-2024 dataset in the CDR vector database, ensuring that the lengths of the retrieved sequences match that of the query. Finally, we proceed to sample the amino acids for each position in the CDR sequences to generate the predicted result for the query CDR. Following ProteinMPNN, we generate two samples for each query and select the one that exhibits the better alignment with the ground-truth as the final result. Average *amino acid recovery (AAR)* is utilized to evaluate model performance, which quantifies the accuracy of the predicted sequences. For a query CDR $q$, the AAR is defined as the ratio of overlapping positions between the predicted sequence $\hat{s}_q$ and ground-truth sequence $s_q$: $\mathrm{AAR}\left(\hat{s}_q, s_q\right) = \frac{1}{L} \sum_{l=1}^{L} \mathbb{I}(\hat{s}_q(l), s_q(l))$.

Fig. 3 (a) illustrates the average AAR for each model on the SAbDab-2024 dataset. As we can observe, Antifold and AbMPNN achieve much better results compared to ProteinMPNN and ESM-IF1, highlighting the advantages of fine-tuning pre-trained protein design models specifically on the antibody dataset. Additionally, IgSeek outperforms its competitors by at least 2.9% on light chain CDR loops (CDR-L) and achieves results comparable to state-of-the-art methods on heavy chain CDR loops (CDR-H). Notably, IgSeek+Kabsch consistently outperforms all baselines across six types of CDR loops, highlighting the effectiveness of our retrieval-based strategy. The marked advantage of IgSeek+Kabsch on CDR-H3 loops is particularly noteworthy, as this type of CDR loops is often considered one of the most hypervariable regions.

**Remark.** We observe a performance degradation in AntiFold and AbMPNN on the SAbDab-2024 dataset compared to the results reported by Høie et al. (2024). One possible reason for this discrepancy is that these two models heavily depend on antibody backbone structures as auxiliary information, while only the structures of CDRs are given in our settings.

**Generalization Performance.** Next, we evaluate the model inference performance on the STCRDab dataset without any further model training. To conduct this evaluation, we randomly draw around 80% of the CDR loops to generate selection templates, while the remaining 20% are used as queries. Fig. 3 (b) displays the average AAR of each model on the STCRDab dataset. As we can see, IgSeek takes the lead by at least 30% on CDR loops from chain A and chain B, respectively. These impressive results further underscore the potential of structure retrieval approaches in mitigating hallucinations during sequence inference, demonstrating that IgSeek can effectively generalize to unseen data while maintaining high accuracy in sequence recovery.

**Efficiency Evaluation.** We evaluate the model efficiency using the OAS-H3 dataset. Fig. 3 (c) reports the inference time of IgSeek compared with other baseline models, all without any model retraining. As we can observe, IgSeek achieves at least 20x speed-up compared to baseline methods, which demonstrates that our IgSeek achieves a better trade-off between effectiveness and efficiency. This enhanced inference speed is particularly beneficial in practical applications like high-throughput antibody design where rapid sequence generation is crucial.

## 5.4 VISUALIZATION

To investigate the representation generated by MEGNN, we conduct a visualization analysis on the SAbDab-before-2024 dataset by T-SNE (Van der Maaten & Hinton, 2008).

Fig. 5 presents the visualization results of top-60 CDR representations in each cluster, where PyIg-Classify cluster labels (Adolf-Bryfogle et al., 2015) (refer to Appendix B) are utilized in this set of experiments. As Fig. 5 illustrates, IgSeek produces a high-quality visualization that clearly organizes the embeddings of CDR loops from distinct clusters into separate groups with minimal overlaps. Furthermore, the visualization not only demonstrates the effectiveness of IgSeek in distinguishing CDRs among different clusters but also highlights its ability to capture structural information inherent in CDR loops. This visual clarity and distinct grouping underscore the robustness and discriminative capability of IgSeek in embedding isomorphic CDR structures closer together while ensuring distinct clusters remain well-separated, which facilitates the identification and retrieval of CDR loops based on their structural characteristics.

## 5.5 CASE STUDY: PDB 8W8R

In this section, we use the 8W8R CDR-L1 as an example to illustrate the query and generation process of IgSeek. Step 1: given the backbone structure of the 8W8R CDR-L1 loop, we employ the pre-trained MEGNN to generate its embeddings. Step 2: we retrieve the top-10 nearest neighbors of the 8W8R CDR-L1 loop from the CDR vector database $\mathcal{Z}$. Step 3: we utilize the aligned sequences from the retrieved records to generate the residue probability distribution at each position. Step 4: Finally, we sample the output result from this distribution. In this example, we observe that the AAR of the sequence generated by IgSeek outperforms other competitors by at least 0.27, demonstrating the effectiveness of our approach.

## 6 CONCLUSION

In this paper, we propose an antibody sequence framework, IgSeek, from a new learning-based structure retrieval perspective. Specifically, IgSeek first constructs a CDR vector database using a multi-channel equivariant graph neural networks. It then predicts CDR sequences from templates retrieved from isomorphic structures in the database. Extensive experiments demonstrate the effectiveness and efficiency of IgSeek, providing insights into de novo antibody sequence design and can inspire further investigatino in this direction.

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

# APPENDIX

## A ANTIBODY

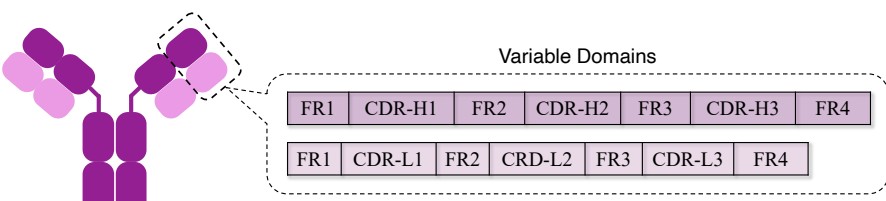

Figure 6: Antibody Structure

Protein is composed by one or multiple chains of amino acid residues which can be twenty different types. An antibody is a special Y-shaped protein with two identical sets of chains, as illustrated in Fig. 6. Each set contains a heavy chain and a light chain, and both of them consist of segments of constant regions and variable regions. The constant regions keep relatively consistent across different antibodies, whereas the variable regions are different to provide different binding regarding the antigen epitope. The variable domains are further separated into alternating fragments of four *framework regions (FRs)* and three *complementarity determining regions (CDRs)*. The CDRs play critical roles in antibody-antigen binding, which is the focus of antibody design.

## B DATASETS AND LABELS

**Datasets.** We selected all experimentally solved antibody structures released in the SAbDab antibody database (Dunbar et al., 2013; Schneider et al., 2021) before January 1, 2024, to sample our training set. We remove CDR sequences that are identical to those in the dataset to eliminate redundancy in the dataset. Following FoldSeek (Van Kempen et al., 2024), for each CDR in the SAbDab-before-2024 dataset, we randomly sample equal-length CDRs with TM-score large than 0.6 to generate training pairs. The final training set consisted of 45, 043 antibody CDR pairs. After

Table 2: Profile of Datasets

| SAbDab | #CDR-H1 | #CDR-H2 | #CDR-H3 | #CDR-L1 | #CDR-L2 | #CDR-L3 |
|---|---|---|---|---|---|---|
| # Data (before-2024) | 4,464 | 4,466 | 4,463 | 3,693 | 3,696 | 3,897 |
| # Query (2024) | 809 | 823 | 513 | 580 | 607 | 578 |
| STCRDab | #CDR-A1 | #CDR-A2 | #CDR-A3 | #CDR-B1 | #CDR-B2 | #CDR-B3 |
| # Data | 680 | 680 | 680 | 741 | 741 | 741 |
| # Query | 138 | 140 | 120 | 158 | 154 | 138 |

finishing model training, all $24,479$ unique CDR structures in the SAbDab-before-2024 dataset are utilized to construct the CDR vector database. The test set of SAbDab-2024 include experimentally solved antibody released in SAbDab antibody database between January 3, 2024 and May 29, 2024. This process resulted in $4,449$ test CDR samples that are completely unseen during the model training process. The sequence similarity distribution between the training set and test set is illustrated in Figure 7. As we can observe, the average sequence similarity for each CDR region in the training and test set is around 0.3 to 0.5, which shows that there is no potential data leakage issue in this data split strategy. In addition, we utilize a T-cell receptor dataset released in the structural T-cell receptor database (Leem et al., 2018) to construct a test set with $5,111$ receptors, referred to as STCRDab. To evaluate the model efficiency, we utilize $5,000$ predicted CDR-H3 loops from the Observed Antibody Space (OAS) (Olsen et al., 2022), denoted as OAS-H3. Redundant CDR loops are removed from the test set. Statistics of these datasets are listed in Table 2.

**Labels.** PyIgClassify cluster labels (North et al., 2011; Adolf-Bryfogle et al., 2015) are employed as ground-truth labels to assess the retrieval performance of antibody CDR regions. For each PDB structure containing an identified antibody heavy or light chain, PyIgClassify categorizes the conformations of CDRs using a three-tier strategy: chain and position, length, and the similarity of dihedral angles. For instance, the cluster ID L1-10-1 denotes a CDR-L1 with a length of 10 amino acids, where the subcluster 1 is determined based on the similarity of dihedral angles using the affinity propagation clustering method (Frey & Dueck, 2007).

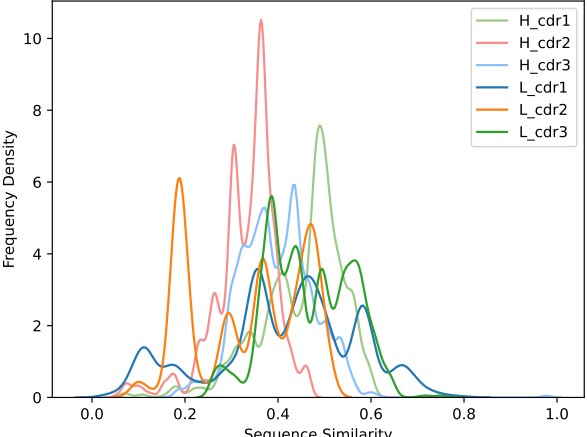

Figure 7: Sequence similarity between SAbDab train/test set.

## C  PROOFS OF THEOREM 1

In this section, we prove that our MEGNN is E(3) equivariant on coordinate $\boldsymbol{X}$ and E(3) invariant on representations $\boldsymbol{h}$ for any transformation operation $g \in E(3)$, more formally:

$$\boldsymbol{h}_i, T_{\mathcal{Y}}(g)\boldsymbol{X}_i^{(L)} = \text{MEGNN}\left(\boldsymbol{h}_i^{(0)}, T_{\mathcal{X}}(g)\boldsymbol{X}_i^{(0)}, G\right).$$

Table 3: Hyperparameters of IgSeek.

| Hyperparameter | Value | Description |
| --- | --- | --- |
| Input | | |
| noise_ratio | 0.15 | Ratio of the input coordinates with added Gaussian noise. |
| noise_scale | 1 | The standard deviation $\sigma$ in the Gaussian noise. |
| $\theta$ | 10 Å | The Euclidean distance threshold when constructing the graph $G$. |
| MEGNN | | |
| learning_rate | $5 \times 10^{-3}$ | Learning rate of MEGNN. |
| weight_decay | $1 \times 10^{-4}$ | Weight decay factor of the optimizer. |
| hidden_dim | 256 | Size of hidden feature dimension in MEGNN. |
| emb_dim | 128 | Size of output embedding dimension in MEGNN. |
| n_layer | 4 | Number of layers in MEGNN. |
| epoch | 50 | Number of the iterations during training |
| batch_size | 8 | Number of batch size in MEGNN. |
| drop_out | 0.1 | Number of dropout rate in MEGNN. |
| Retrieval | | |
| $k$ | 10 | Number of nearest neighbor retrieved in the CDR vector database. |
| n_sample | 2 | Number of generated samples for each query. |

*Proof.* We assume that $h_i^{(0)}$ is invariant to E(3) transformation operations on the coordinate $X_i^{(0)}$, since $h_i^{(0)}$ is generated from uniform distribution and no absolute information of $X_i^{(0)}$ is encoded into $h_i^{(0)}$. Then, for the E(3) transformation $g := RX + b$, where orthogonal matrix $R \in O(3)$ and bias $b \in \mathbb{R}^3$, we have:

$$RX_i^{(l-1)} + b - (RX_j^{(l-1)} + b) = RX_{ij}^{(l-1)},$$

$$(RX_{ij}^{(l-1)})^\top RX_{ij}^{(l-1)} = z_{ij}^{(l-1)}.$$

Therefore, the output $z_{ij}^{(l-1)}$ of Eq. 5 is E(3) invariant to transformation $g$.

As for Eq. 6, since $h_i$, $h_j$, and $z_{ij}^{(l-1)}$ are invariant to E(3) transformation operations, we can derive that $h_{e_{ij}}^{(l)}$ is E(3) invariant.

Next, we will prove Eq. 7 is E(3) equivariant.

$$RX_i^{(l-1)} + b + \frac{1}{|\mathcal{N}_i|} \sum_{v_j \in \mathcal{N}_i} \text{MLP}\left(h_{e_{ij}}^{(l)}\right) \cdot RX_{ij}^{(l-1)} = R\left(X_i^{(l-1)} + \frac{1}{|\mathcal{N}_i|} \sum_{v_j \in \mathcal{N}_i} m_{j \to i}\right) + b$$

$$= RX_i^{(l)} + b.$$

Therefore, we have proven that any E(3) transformation operations on $X_i^{(l-1)}$ leads to the same E(3) transformation operations on $X_i^{(l)}$ using Eq. 7.

Finally, it is easy to verify that Eq. 8 is E(3) invariant as $h_i^{(l-1)}$ and $h_{e_{ij}}^{(l)}$ are E(3) invariant.

In conclusion, for an $L$-layer MEGNN model, any transformation $g \in E(3)$ on the input coordinate $X^{(0)}$ will lead to the same E(3) transformation operations on the output coordinate $X^{(L)}$ while the representations $h^{(L)}$ still remain E(3) invariant:

$$h_i, T_{\mathcal{Y}}(g)X_i^{(L)} = \text{MEGNN}\left(h_i^{(0)}, T_{\mathcal{X}}(g)X_i^{(0)}, G\right).$$

This finishes the proof. $\qquad\square$

# D  IMPLEMENTATION DETAILS

In this section, we introduce the implementation details of our IgSeek. The MEGNN model introduced in Section 4 consists of three key learnable functions:

---

**Algorithm 1:** Multi-channel Equivariant Graph Neural Network (MEGNN)

---

**Input:** Antibody CDR Structure $G = (V, E)$, initial features $\boldsymbol{h}_i^{(0)}$ and coordinates $\boldsymbol{X}_i^{(0)}$ for each node $v_i \in V$

**Output:** Antibody CDR representation $\boldsymbol{h}_G$

1 Initialize coordinates $\hat{\boldsymbol{X}}_i \leftarrow \boldsymbol{X}_i + \mathcal{N}(0, \sigma)$;

2 **for** *layer $l = 1$ to $L$* **do**

3    **for** $v_i \in V$ **do**

4       **for** $v_j \in \mathcal{N}_i$ **do**

5          Calculate the coordinate differences: $\boldsymbol{X}_{ij}^{(l-1)} \leftarrow \boldsymbol{X}_i^{(l-1)} - \boldsymbol{X}_j^{(l-1)}$ ;

6          Calculate the square distance: $\boldsymbol{z}_{ij}^{(l-1)} \leftarrow (\boldsymbol{X}_{ij}^{(l-1)})^\top \boldsymbol{X}_{ij}^{(l-1)}$ ;

7          Update the edge feature: $\boldsymbol{h}_{e_{ij}}^{(l)} \leftarrow \phi_e \left( \text{CONCAT} \left( \boldsymbol{h}_i^{(l-1)}, \boldsymbol{h}_j^{(l-1)}, \boldsymbol{z}_{ij}^{(l-1)} \right) \right)$ ;

8          Derive the propagated information: $\boldsymbol{m}_{j \leftarrow i} \leftarrow \text{MLP} \left( \boldsymbol{h}_{e_{ij}}^{(l)} \right) \cdot \boldsymbol{X}_{ij}^{(l-1)}$ ;

9       Update the coordinate: $\boldsymbol{X}_i^{(l)} \leftarrow \boldsymbol{X}_i^{(l-1)} + \frac{1}{|\mathcal{N}_i|} \sum_{v_j \in \mathcal{N}_i} \boldsymbol{m}_{j \to i}$;

10       Derive the aggregated edge feature: $\boldsymbol{h}_{agg_i}^{(l)} \leftarrow \sum_{j \in \mathcal{N}_i} \boldsymbol{h}_{e_{ij}}^{(l)}$ ;

11       Update the node representation: $\boldsymbol{h}_i^{(l)} \leftarrow \boldsymbol{h}_i^{(l-1)} + \text{MLP} \left( \text{CONCAT}(\boldsymbol{h}_i^{(l-1)}, \boldsymbol{h}_{agg_i}^{(l)}) \right)$ ;

12 Generate the representation of input CDR structure: $\boldsymbol{h}_G \leftarrow \text{READOUT}(\{\boldsymbol{h}_i^{(L)}\}_{i=1}^n)$ ;

13 Return $\boldsymbol{h}_G$;

---

- The edge module $\phi_e$ (refer to Eq. 6) consists of a two-layer MLP with two Leaky Rectified Linear Unit (LeakyReLU) activation functions (Xu et al., 2015). Besides, a dropout function (Srivastava et al., 2014) with $0.1$ dropout rate is employed on the output of $\phi_e$:

  CONCAT(Features) $\to$ Input $\to$ {LinearLayer() $\to$ LeakyReLU() $\to$ LinearLayer()
  
  $\to$ LeakyReLU()} $\to$ Dropout $\to$ Output.

- The coordinate module $\phi_X$ (refer to Eq. 7) contains a two-layer MLP that shares weights with the MLP in the edge module $\phi_e$.

- The node module $\phi_h$ (refer to Eq. 8) is a two-layer MLP with one LeakyReLU activation function:

  CONCAT(Features) $\to$ Input $\to$ {LinearLayer() $\to$ LeakyReLU() $\to$ LinearLayer()} $\to$ Output.

In our experiments, we train the MEGNN model in IgSeek using PyTorch (Paszke et al., 2019) with an Adam optimizer (Kingma & Ba, 2015) on 4 NVIDIA Tesla A100 GPUs. Table 3 lists the hyperparameters of IgSeek.

## E   ALGORITHM

In Section 4, we provide a comprehensive overview of the IgSeek framework. We first introduce the MEGNN encoder in Section 4.1, followed by a discussion of the MEGNN decoder and the sequence prediction process utilized by IgSeek. Here, we provide the algorithms of IgSeek as complementary details. Specifically, Algorithm 1 summarizes the forward pass of MEGNN, Algorithm 2 outlines the training process, and Algorithm 3 presents the antibody CDR sequence design process, respectively.

## F   BASELINES

The first category is structure retrieval model:

- **FoldSeek** (Van Kempen et al., 2024) represents tertiary amino acid interactions using 3D interaction (3Di) structural alphabet, achieving 4 to 5 orders of magnitude speed-up compared to traditional iterative or stochastic structure retrieval methods like CE (Shindyalov & Bourne, 1998), Dali (Holm, 2020), and TM-align (Zhang & Skolnick, 2005). Official code is available at: https://github.com/steineggerlab/foldseek.

---

**Algorithm 2:** CDR Vector Database Construction

---

**Input:** Training set $\mathcal{T} = \{(G_{i1}, G_{i2})\}_{i=1}^{|\mathcal{T}|}$, training epoch $T$, CDR database $\mathcal{B} = \{(\boldsymbol{s}_j, G_j)\}_{j=1}^{|\mathcal{B}|}$
**Output:** CDR vector database $\mathcal{Z}$

1 **for** $t = 1$ *to* $T$ **do**
2    **for** $i = 1$ *to* $|\mathcal{T}|$ **do**
3      Initialize feature matrices $\boldsymbol{H}_{i1}$ of $G_{i1}$ and feature matrices $\boldsymbol{H}_{i2}$ of $G_{i2}$, respectively
4      Generate graph representation for the $i$-th training CDR pair:
       $h_{G_{i1}} \leftarrow \text{MEGNN}(G_{i1}, \boldsymbol{H}_{i1}, \boldsymbol{X}_{i1}), \quad h_{G_{i2}} \leftarrow \text{MEGNN}(G_{i2}, \boldsymbol{H}_{i2}, \boldsymbol{X}_{i2})$
5      Predict the RMSD between $h_{G_{i1}}$ and $h_{G_{i2}}$:
       $\widehat{d}(G_{i1}, G_{i2}) \leftarrow \text{MLP}(\text{CONCAT}(\boldsymbol{h}_{G_{i1}}, \boldsymbol{h}_{G_{i2}}))$
6      Compute the loss function: $\mathcal{L} \leftarrow \frac{1}{|\mathcal{T}|} \sum_{(G_{i1}, G_{i2}) \in \mathcal{T}} \|\widehat{d}(G_{i1}, G_{i2}) - d(G_{i1}, G_{i2})\|^2$
7    Update the model weights $\boldsymbol{W}$ to minimize $\mathcal{L}$ using $\frac{\partial \mathcal{L}}{\partial \boldsymbol{W}}$
8 **for** $j = 1$ *to* $|\mathcal{B}|$ **do**
9    Generate graph representation $G_j \leftarrow \text{MEGNN}(G_j, \boldsymbol{H}_j, \boldsymbol{X}_j)$
10    Add the triplet $(\boldsymbol{s}_j, G_j, \boldsymbol{h}_{G_j})$ into $\mathcal{Z}$
11 Return vector database $\mathcal{Z}$

---

**Algorithm 3:** Sequence Generation

---

**Input:** Query structure $G_q$, MEGNN $\phi$, CDR vector database $\mathcal{Z}$
**Output:** Predicted sequence $\hat{\boldsymbol{s}}_q$

1 Initialize feature matrix $\boldsymbol{H}_q$ and coordinates $\boldsymbol{X}_q$
2 Generate graph representation $G_q \leftarrow \text{MEGNN}(G_q, \boldsymbol{H}_q, \boldsymbol{X}_q)$
3 Retrieve the $K$-nearest neighbors of $\boldsymbol{h}_G$ in the database $\mathcal{Z}$ as $\mathcal{Z}_q$
4 Derive the probability of amino acid $a$ at the $l$-th position:
   $p(\hat{\boldsymbol{s}}_q(l) = a_i | S_q) = \frac{1}{K} \sum_{s_k \in \mathcal{S}_q} \mathbb{I}(\boldsymbol{s}_k(l), a_i)$
5 Sample the amino acid $\hat{\boldsymbol{s}}_q(l)$ at the $l$-th position using the probability $p(\hat{\boldsymbol{s}}_q(l) | \mathcal{S}_q)$
6 Return $\hat{\boldsymbol{s}}_q$

---

The second category is protein and antibody design models:

- **ProteinMPNN** (Dauparas et al., 2022) is a deep learning–based method for protein sequence design that excels in both in silico and experimental evaluations, achieving a sequence recovery of 52.4% on native protein backbones, compared to 32.9% for Rosetta (Adolf-Bryfogle et al., 2018; Baek et al., 2021). By leveraging a message-passing neural network with enhanced input features and edge updates, ProteinMPNN is capable of designing monomers, cyclic oligomers, protein nanoparticles, and protein-protein interfaces, rescuing previously failed designs generated by Rosetta (Baek et al., 2021) or AlphaFold (Jumper et al., 2021). Official code is available at: https://github.com/dauparas/ProteinMPNN.

- **ESM-IF1** (Hsu et al., 2022) employs a sequence-to-sequence Transformer to predict protein sequences from backbone atom coordinates, which is pre-trained on structures of 12M protein sequences. It achieves 51% native sequence recovery and 72% for buried residues. Official code is available at: https://github.com/facebookresearch/esm/tree/main/examples/inverse_folding.

- **AbMPNN** (Dreyer et al., 2023) fine-tunes ProteinMPNN on the SAbDab (Dunbar et al., 2013; Schneider et al., 2021) dataset for antibody design, outperforming generic protein models in sequence recovery and structure robustness, especially for the hypervariable CDR-H3 loop. The profile of model weights is available at: https://zenodo.org/records/8164693.

- **AntiFold** (Høie et al., 2024) is an antibody-specific inverse folding model fine-tuned from ESM-IF1 (Hsu et al., 2022) on solved antibody structures from the SAbDab dataset (Dunbar et al., 2013; Schneider et al., 2021) and predicted antibody structures from the OAS dataset (Kovaltsuk et al., 2018; Olsen et al., 2022). AntiFold excels in sequence recovery and structural similarity while also demonstrates stronger correlations in predicting antibody-antigen binding affinity in a zero-shot manner. Official code is available at: https://github.com/oxpig/AntiFold.

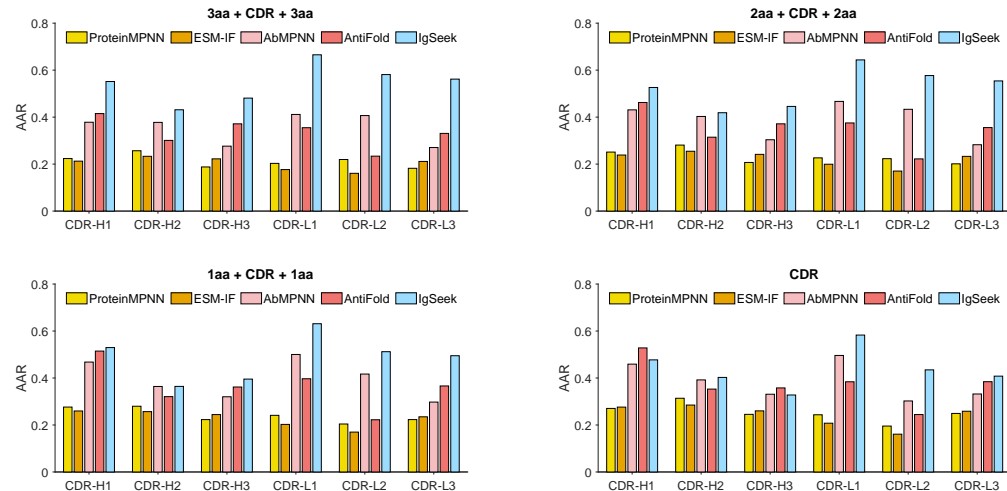

Figure 8: The Comparison of Average AAR on the SAbDab-2024 Dataset using CDRs with extensions of 1 to 3 amino acids on each side in the flanking regions.

Table 4: The Comparison of Average AAR with varying $K$ in SAbDab-2024.

| $K$ | 5 | 10 | 20 | 50 | 100 |
|---|---|---|---|---|---|
| CDR-L1 | 0.660 | 0.658 | 0.645 | 0.620 | 0.593 |
| CDR-L2 | 0.580 | 0.580 | 0.573 | 0.573 | 0.550 |
| CDR-L3 | 0.586 | 0.586 | 0.576 | 0.574 | 0.564 |
| CDR-H1 | 0.560 | 0.561 | 0.560 | 0.553 | 0.537 |
| CDR-H2 | 0.440 | 0.435 | 0.432 | 0.429 | 0.430 |
| CDR-H3 | 0.473 | 0.464 | 0.455 | 0.447 | 0.441 |

## G  ADDITIONAL EXPERIMENTS

**CDR with extensions.** In this set of experiments, we compare IgSeek with protein and antibody design baselines using the SAbDab-2024 dataset. We focus on CDRs with backbone extensions of $n$ amino acids on each side in the flanking regions. Fig. 8 illustrates the results for varying values of $n = 0, 1, 2, 3$. As we can observe, the performance of IgSeek improves with the inclusion of additional amino acids in the given structure, , which aligns with the fact that more input structural information can be encoded into the CDR representation. In contrast, other baseline models are adversely affected by hallucinations stemming from conserved backbone structures. Notably, when $n = 3$, IgSeek consistently outperforms its competitors by at least $5\%$ and $18\%$ for heavy chain and light chain CDR loops, respectively. This further demonstrates that the retrieval-based strategy employed by IgSeek effectively mitigates hallucinations during CDR sequence generation.

**Influence of value $K$.** In this set of experiments, we conduct experiments on the SAbDab-2024 dataset to evaluate the impact of varying parameter $K$ in IgSeek. Table 4 reports the average AAR of IgSeek across different values of $K$ on the SAbDab-2024 dataset. As we can observe, the performance of IgSeek exhibits a decline as $K$ increases. In our implementation, we set $K = 10$ rather than 5 as IgSeek achieves comparable results while preserving enhanced sequence diversity.

