# OpenReview forum: "IgSeek: Fast and Accurate Antibody Design via Structure Retrieval"
_ICLR.cc/2025/Conference — Submitted to ICLR 2025_

### Official Review · Reviewer_9YZG · 2024-10-18

**Soundness:** 2
**Presentation:** 3
**Contribution:** 2
**Rating:** 5
**Confidence:** 5

**Summary:**

The design of synthetic antibodies is an important challenge in AI for computational biology. This paper notices the occurrence of hallucinations during sequence inference, where sequences may not fold into desired structures in real applications. To overcome this obstacle, IgSeek proposes to retrieve similar structures from a natural antibody database. Experiments demonstrate its high efficiency by achieving the state-of-the-art performance in sequence recovery. Despite the performance, there are several issues remained to be explained more clearly.

**Strengths:**

(1) The idea of searching and matching substructures from a given candidate database is novel and interesting. I believe sometimes, instead of completely de novo generation, retrieval-based algorithm is a promising direction and can be incorporated into de novo design.

(2) The inference speed of IgSeek is outstanding, which may enable the large-scale antibody retrieval-based design.

**Weaknesses:**

(1) The multi-channel equivariant message passing (MEGNN) has already been introduced in MEAN [i]. What is the difference between this paper's MEGNN and MEAN's version? They look very similar, just extending the single-channel EGNN layer to process residues with several atom coordinates.


[i] CONDITIONAL ANTIBODY DESIGN AS 3D EQUIVARIANT GRAPH TRANSLATION. ICLR 2023.

(2) Some important baselines are missing, for instance, DiffAB, MEAN, dyMEAN, etc.

(3) A key motivation for IgSeek is that prior DL methods such as ProteinMPNN and IF can trigger hallucinations, namely, inferred sequences fail to fold into expected structures. *Therefore, it can be biased if we only adopt the sequence recovery rate to evaluate the model performance.*

However, in the experiments, the author still regarded AAR as the most crucial factors to assess the performance of different approaches. From my point of view, they should compare the designed CDR structures (via some antibody-antigen folding prediction models such as Ab-Fold, AF3, etc.) with the ground truth structures. Besides, it is also necessary to evaluate the binding affinity or energy change between designed CDR and the original one, which can be difficult and can be implemented using some simulation tools such as Rosetta. My major concern is that you can not take the retrieved CDR struture for granted as the real one for designed CDR. Even if with the same sequences, these substructures can be significantly different in different antibody contexts. However, if the author have literactures or theories to support their point, I would be happy to discuss further.

**Questions:**

(1) Unlike previous co-design methods, IgSeek only focus on the CDR, ignoring the other parts of the antibody and even antigens. The current task in this paper is more like a toy task, without considering all essential contexts and ultimate goals: given an antigen, we hope to design an optimal antibody that binds well with it. So from both sides of model design and evaluation, we should all take this factor into account. Can the author explain why they only care about CDR and believe a successful CDR match is adequate for antibody design?


(2) The experimental results are heavily dependent on your split and curation of the dataset. In other words, if there are no similar CDRs in your CDR candidate databse, you can never find or retrieve a suitable CDR for test samples. Unfortunately, the author failed to explore this severe drawbacks and proposed any potential solutions.

---

> ### Author Response · Authors · 2024-11-26
> **Author Response (1/2)**
>
> We appreciate your comprehensive and constructive review. Our point-to-point responses to your comments are given below.
>
> ---
>
> **W1** The multi-channel equivariant message passing (MEGNN) has already been introduced in MEAN. What is the difference between this paper's MEGNN and MEAN's version? They look very similar, just extending the single-channel EGNN layer to process residues with several atom coordinates.
>
> **RW1**. Please kindly refer to **RW3** to Reviewer **tHJL**.
>
> ---
>
> **W2** Some important baselines are missing, for instance, DiffAB, MEAN, dyMEAN, etc.
>
>
> **RW2**: Please kindly refer to **RW7** to Reviewer **tHJL**.
>
> ---
>
> **W3** A key motivation for IgSeek is that prior DL methods such as ProteinMPNN and IF can trigger hallucinations, namely, inferred sequences fail to fold into expected structures. Therefore, it can be biased if we only adopt the sequence recovery rate to evaluate the model performance. However, in the experiments, the author still regarded AAR as the most crucial factors to assess the performance of different approaches. From my point of view, they should compare the designed CDR structures (via some antibody-antigen folding prediction models such as Ab-Fold, AF3, etc.) with the ground truth structures. Besides, it is also necessary to evaluate the binding affinity or energy change between designed CDR and the original one, which can be difficult and can be implemented using some simulation tools such as Rosetta.
>
> **RW3**. Thank you for your insightful feedback. We agree that a comprehensive evaluation of antibody design should encompass not only amino acid recovery (AAR) but also metrics such as root-mean-square deviation (RMSD) and binding free energy (ΔΔG). Due to time constraints, we plan to include these evaluations in our future work.
>
> Regarding RMSD evaluation, we anticipate that IgSeek will perform well since it retrieves CDRs from a natural antibody database rather than relying on predicted antibody CDR structures. For binding energy evaluation, the current model lacks antigen epitope information, which is crucial for accurate assessment. To address this, we may need to substitute the contact residues in the CDR, known as paratopes, with chemically complementary ones, either through the expertise of experienced antibody engineers or by incorporating antigen epitope information into the model for future chemical complementarity reasoning. For more information, please kindly refer to **RW2** to Reviewer **MRmb**.
>
> **W4** My major concern is that you cannot take the retrieved CDR structure for granted as the real one for designed CDR. Even if with the same sequences, these substructures can be significantly different in different antibody contexts. However, if the authors have literatures or theories to support their point, I would be happy to discuss further.
>
>
> **RW4**. Thank you for your thorough and insightful feedback. One valuable piece of literature that enlightened our study is Bennett et al. [1], recently posted on bioRxiv this year. In response to your major concern, we would like to clarify the original motivation of our study. Following Bennett et al., we believe that a pharmaceutically practical AI model for designing therapeutic antibodies should generate CDR structures and sequences onto a unified antibody framework to target different antigens. Our work aims to enhance the accuracy of CDR sequences given their structures through natural CDR retrieval. The generation of CDR structures could be achieved by RFdiffusion [2], as done in Bennett et al., or other generative models. Alternatively, our model is highly compatible with integration into a structure generative model to achieve accurate CDR generation augmented by structural retrieval, where IgSeek plays a role similar to multiple sequence alignment in protein structure prediction, an inverse question of protein design. For more information, please kindly refer to **RW2** to Reviewer **MRmb** and **RW5** to Reviewer **tHJL**.

---

> ### Author Response · Authors · 2024-11-26
> **Author Comment (2/2)**
>
> ---
>
> **Q1**. Unlike previous co-design methods, IgSeek only focus on the CDR, ignoring the other parts of the antibody and even antigens. The current task in this paper is more like a toy task, without considering all essential contexts and ultimate goals: given an antigen, we hope to design an optimal antibody that binds well with it. So from both sides of model design and evaluation, we should all take this factor into account. Can the author explain why they only care about CDR and believe a successful CDR match is adequate for antibody design?
>
>
> **RQ1**. Please kindly refer to **RW3** and **RW4**.
>
> ---
>
> **Q2**. The experimental results are heavily dependent on your split and curation of the dataset. In other words, if there are no similar CDRs in your CDR candidate database, you can never find or retrieve a suitable CDR for test samples. Unfortunately, the author failed to explore these severe drawbacks and proposed any potential solutions.
>
> **RQ2**. Please kindly refer to **RW9** to Reviewer **L9vp**.
>
> ---
>
> Reference:
>
> [1] N. R. Bennett, et al. "Atomically accurate de novo design of single-domain antibodies." 2024, bioRxiv.
>
> [2] J. L. Watson, et al. "De novo design of protein structure and function with RFdiffusion." Nature 620(7976): 1089-1100, 2023.
>
> ---
>
> We sincerely appreciate your time, and we are glad to answer any additional questions you may have.

---

> > ### Comment · Reviewer_9YZG · 2024-11-26
> >
> > As the reviewer who gave the highest score among all four reviewers, I feel bad when I saw the response is filled with **Please kindly refer to RWxxx to Reviewer xxx**. It took a lot of extra time to go back and forth to read other reviewers' comments and the so-called "RWxxx". I tend to be a responsible reviewer and am glad to see that many of my questions are also jointly asked by other reviewers. However, I believe it would not take much time for the authors just to copy and paste their answers here instead of asking me to refer to somewhere else again and again.

---

### Official Review · Reviewer_tHJL · 2024-11-03

**Soundness:** 3
**Presentation:** 3
**Contribution:** 2
**Rating:** 3
**Confidence:** 4

**Summary:**

This paper presents IgSeek, a novel structure-retrieval framework for antibody design. The framework leverages neural retrieval in an antibody database to retrieve structurally similar sequence templates of Complementarity-Determining Regions (CDRs) and ensembles these templates for sequence prediction. The paper demonstrates the effectiveness of IgSeek in predicting CDR sequences, particularly for relatively conserved residues, and shows its superiority in terms of speed and accuracy compared to existing baseline models.

**Strengths:**

1. IgSeek introduces a novel structure-retrieval framework for antibody design, which leverages neural retrieval and ensembling techniques.
2. The experimental setup is rigorous, and the results are thoroughly analyzed and visualized.

**Weaknesses:**

1. What is definition of the "hallucinations" problem in antibody design? How this problem generated?
2.  The authors said that the "IgSeek leverages neural retrieval in an antibody database to retrieve structurally similar sequence templates of CDR". I guess the performance is dependent on the accuracy of the antibody structures, How to get accurate antibody structures, how to ensure that the antibody database is large, and what if no structurally similar samples can be retrieved for some antibodies?
3. Other than the structure retrieval, what are the differences of your GNN method with other similar GNN-based antibody design methods?
4.  Some notations should be specified,  what does the $x_{ij}$ mean in Eq.4, the K value in Line 99, why k is 10 as shown in Table 2, how to determine it?
5. IgSeek needs the CDR structures as input, this may limit the application of this method, how to design the antibodies when the CDR structures are not known?  The effectiveness of IgSeek is heavily dependent on the quality and diversity of the antibody database. How sensitive is IgSeek to variations in the antibody database, and what strategies can be employed to mitigate this sensitivity?
6. it seems that there is no specific design for antibody, this method is very general, i.e., it is can be used in common protein design. Structure retrieval method can be used in other structure to sequence tasks.
7. Lack of comparison baselines in Figure 3, like MEAN (Kongetal.,2023b), dyMEAN(Kongetal., 2023a) and ADesigner (tan, 2023).


[1] Xiangzhe Kong,et al. End-to-end full-atom antibody design. In ICML, pp.17409–17429,2023a.
[2] Xiangzhe Kong,et al. Conditional antibody design as 3d equivariant graph translation. In ICLR,2023b.
[3] Tan, C., et al. Cross-gate mlp with protein complex invariant embedding is a one-shot antibody designer. In AAAI.

**Questions:**

1. Can IgSeek be adapted to design antibodies for specific targets or diseases? If so, what modifications would be required?
2. Why diffusion models can be used to design antibodies?

**Details Of Ethics Concerns:**

Limitations and potential influence are not presented in this manuscript. Besides, this paper is longer than 9 pages, I am not whether this is allowed or not.

---

> ### Author Response · Authors · 2024-11-26
> **Author Response (1/2)**
>
> We appreciate your comprehensive and constructive review. Our point-to-point responses to your comments are given below.
>
> ---
>
> **W1**. What is definition of the "hallucinations" problem in antibody design? How this problem generated?
>
> **RW1**. That is a good question. We apologize for not clearly defining the term "hallucination" in the context of antibody design in our manuscript. In general, hallucination refers to the phenomenon where large language models (LLMs) generate responses that sound plausible but are factually incorrect or entirely fabricated. In the field of immunology, antibody generation is an immune response (answer) to an external antigen invasion (question). Similarly, in computational antibody design, we define hallucination as the situation where a model, given an antigen epitope and a CDR backbone, generates CDR sequences that appear structurally fitting (as predicted by AlphaFold2, for example) but are actually non-functional, i.e., they do not fold correctly or do not bind to the antigen.
>
> In our Introduction section, we proposed two potential reasons for hallucination in computational antibody design: (1) The sequences inferred by the model may not fold into the desired structure in wet lab conditions. (2) Independent structure prediction models may have low confidence in predicting CDR structures due to their high flexibility. Additionally, we would like to supplement two more possible reasons: (1) Unlike LLMs, which are trained on extensive text data from the entire internet, high-quality antigen-antibody paired data are limited and insufficient for training large-scale antibody generation models. (2) Many antibody generative models use an autoregressive approach to generate CDR sequences. However, this does not mimic the in vivo process of antigen generation, where naïve B cells undergo numerous rapid cycles of recombination, mutation, and selection.
>
> ---
>
> **W2**. The authors said that the "IgSeek leverages neural retrieval in an antibody database to retrieve structurally similar sequence templates of CDR". I guess the performance is dependent on the accuracy of the antibody structures, how to get accurate antibody structures, how to ensure that the antibody database is large, and what if no structurally similar samples can be retrieved for some antibodies?
>
> **RW2**. Please refer to **RW7** and **RW9** to Reviewer **L9vp**.
>
> ---
>
> **W3**. Other than the structure retrieval, what are the differences of your GNN method with other similar GNN-based antibody design methods?
>
> **RW3**. Our MEGNN model is uniquely designed to generate embeddings specifically for CDR regions, distinguishing it from other GNN-based antibody design methods. One major difference is that while most existing models address the antibody co-design problem by requiring the input of antibody framework sequences, which are strongly interdependent with CDR1 and CDR2 due to their common coding germline V genes, our model directly targets the desired conformation without needing the antibody framework sequence information. This allows our model to accommodate any given antibody framework with good developability and clinical success to target different antigens. Additionally, unlike other approaches that treat the task as a regression problem and mask the queried CDR amino acids, our model explicitly recovers the highly conserved amino acids that naturally govern the CDR conformations. Furthermore, the loss function in MEGNN is specifically optimized to maintain the RMSD of input pairs, without incorporating other complex features. This targeted approach significantly reduces computational costs compared to previous methods.
>
> ---
>
> **W4** Some notations should be specified, what does the xij mean in Eq.4, the K value in Line 99, why k is 10 as shown in Table 2, how to determine it?
>
>
> **RW4**. $x_{ij}$ is the coordinate differences between node $i$ and node $j$. We have included a new Table 4 for the parameter analysis in Appendix G. As we can observe, the performance of IgSeek exhibits a decline as $K$ increases. In our implementation, we set $K=10$ rather than $5$ as IgSeek achieves comparable results while preserving enhanced sequence diversity.

---

> ### Author Response · Authors · 2024-11-26
> **Author Response (2/2)**
>
> ---
>
> **W5** IgSeek needs the CDR structures as input, this may limit the application of this method, how to design the antibodies when the CDR structures are not known? The effectiveness of IgSeek is heavily dependent on the quality and diversity of the antibody database. How sensitive is IgSeek to variations in the antibody database, and what strategies can be employed to mitigate this sensitivity?
>
> **RW5**. Thank you for your comments. In response to your first concern, the key issue is defining what constitutes truly de novo antibody design in the industry. Bennett et al. [1] have proposed a pharmaceutically practical approach that involves designing CDRs onto a unified framework to target different antigens, which we believe aligns more closely with the needs of the pharmaceutical industry compared to previous methods. Specifically, the CDR conformation can be derived using RFdiffusion [2] (or other generative methods), a diffusion model that generates backbone conformations based on the antigen epitope and the unified antibody framework. Once the CDR conformation is obtained, IgSeek can then be applied to predict the corresponding sequence. We recognize that the effectiveness of this strategy may depend on the quality of the structures generated by the diffusion models, and we plan to explore this aspect in future studies. In this current study, our objective was to evaluate the performance of our method using natural CDR backbone conformations.
>
> Regarding the second question about the sensitivity of IgSeek to variations in the antibody database, one potential solution is to enrich the database with large-scale antibody structures predicted by models such as AlphaFold2 [3]. Due to time constraints, we have not yet implemented this strategy, but it is a key focus of our future work.
>
> ---
>
>
>
> **W6**. it seems that there is no specific design for antibody, this method is very general, i.e., it can be used in common protein design. Structure retrieval method can be used in other structure to sequence tasks.
>
> **RW6**. Yes, you are correct. The structure retrieval method we have developed for antibodies is indeed versatile and can be applied to the design of other types of proteins, such as enzymes.
>
> ---
>
> **W7**. Lack of comparison baselines in Figure 3, like MEAN, dyMEAN, and ADesigner.
>
> **RW7**: Thank you for your valuable feedback. We acknowledge the importance of including comparison baselines in Figure 3. However, it is important to note that there are significant differences in the antibody generation settings between IgSeek and models such as MEAN, dyMEAN, and ADesigner. These models typically require both antigen and antibody frameworks as inputs, whereas our approach focuses solely on CDR conformations. This fundamental difference makes a direct comparison challenging and potentially unfair. To address this concern and provide clarity, we have revised our manuscript to include a new Table 1, which presents a comparative analysis of the different antibody design task configurations. This table aims to highlight the distinct requirements and settings of each model, thereby providing a clearer context for evaluating their performance.
>
> ---
>
> **Q1**. Can IgSeek be adapted to design antibodies for specific targets or diseases? If so, what modifications would be required?
>
> **RQ1**. Please kindly refer to **RW5**.
>
> ---
>
> **Q2**. Why diffusion models can be used to design antibodies?
>
>
> **RQ2**. We apologize for any confusion. We would like to clarify that diffusion models can be utilized to design antibody structures, which can then be input into IgSeek for sequence inference.
>
> ---
>
> Reference:
>
> [1] N. R. Bennett, et al. "Atomically accurate de novo design of single-domain antibodies." 2024, bioRxiv.
>
> [2] J. L. Watson, et al. "De novo design of protein structure and function with RFdiffusion." Nature 620(7976): 1089-1100, 2023.
>
> [3] J. Jumper, et al. "Highly accurate protein structure prediction with AlphaFold." Nature 596(7873): 583-589, 2021.
>
> ---
>
> We sincerely appreciate your time, and we are glad to answer any additional questions you may have.

---

> > ### Comment · Reviewer_tHJL · 2024-11-26
> > **Response by Reviewer**
> >
> > Thanks for your time and effort. This rebuttal is too late. The authors did not answer my questions well, especially the W2 and W7. I will keep my score.

---

### Official Review · Reviewer_L9vp · 2024-11-03

**Soundness:** 2
**Presentation:** 2
**Contribution:** 2
**Rating:** 3
**Confidence:** 4

**Summary:**

The paper presents **IgSeek**, a framework for sequence design of the complementarity-determining regions (CDRs) of antibodies. It addresses challenges in AI-driven protein design, particularly the issue of hallucinations in hyper-variable antibody regions. Instead of direct sequence generation, IgSeek uses a multi-channel equivariant graph neural network to obtain embeddings for CDR loops and retrieves similar structures from an antibody database to infer sequences. The authors claim that IgSeek outperforms existing methods in accuracy and efficiency for both antibody and T-cell receptor sequence recovery tasks although the experimental setup is flawed.

**Strengths:**

- The paper is well-written and easy to follow.
- IgSeek presents a structure-retrieval approach for antibody CDR loops, enabling sequence inference based on a given CDR loop backbone structure. This method is similar to FoldSeek [1], which was developed for efficient structure search in general proteins.
- The framework shows strong performance and fast inference compared to state-of-the-art protein and antibody models. However, the experimental setup is flawed.

**References:** \
[1] Michel Van Kempen, Stephanie S Kim, Charlotte Tumescheit, Milot Mirdita, Jeongjae Lee, Cameron LM Gilchrist, Johannes Soding, and Martin Steinegger. Fast and accurate protein structure search with foldseek. Nature biotechnology, 42(2):243–246, 2024. doi: https://doi.org/10.1038/s41587-023-01773-0.

**Weaknesses:**

- The method lacks novelty, as the clustering of canonical CDR loops has been previously established ([1]) and used for antibody sequence design in frameworks such as Rosetta ([2]).

- The comparison with FoldSeek in Figure-2 is problematic; FoldSeek is designed for conserved regions of general proteins and is not suited for flexible loops such as antibody CDRs. Despite that, it’s surprising to me that FoldSeek outperformed IgSeek in CDR-H3.

- In Figure-3, the comparisons with Antifold and AbMPNN for amino acid recovery tasks are unfair, as these models are not specifically trained for CDR loops alone. As acknowledged by the authors, this biases the results in favour of IgSeek, potentially exaggerating its performance.

- Figure-3 indicates that sorting by RMSD is more effective than using embeddings from IgSeek. Showing results based solely on RMSD (Kabsch) would provide a clearer performance baseline.

- Figure-4 omits the inference time of the IgSeek+Kabsch method, although the authors use this method to argue that their approach outperforms others in Figure-3a/b. The added time from Kabsch alignment likely impacts overall inference speed, and a consistent comparison across methods is important.

- In the CDR sequence recovery tasks, the authors sample two sequences per query and select the one that best aligns with the ground truth, which is biased. In practical applications, such as library design for real experiments, the ground truth is unknown.

- IgSeek’s effectiveness heavily relies on the quality and diversity of the antibody structure database, which may restrict its applicability. This dependency also limits its ability to generate novel, diverse sequences, reducing its utility for de novo design.

- The paper does not specify how CDR regions are defined  (what numbering scheme is used?)

- Generated sequences for a given CDR loop are constrained to a fixed length, limiting flexibility for generating variable-length sequences, although similar limitations apply to existing methods.


**References:** \
[1] North B, Lehmann A, Dunbrack RL Jr. (2011). A new clustering of antibody CDR loop conformations. J Mol Biol 406: 228–256. pmid:21035459 \
[2] Jared Adolf-Bryfogle, Qifang Xu, Benjamin North, Andreas Lehmann, and Roland L Dunbrack Jr. Pyigclassify: a database of antibody cdr structural classifications. Nucleic acids research, 43(D1): D432–D438, 2015. doi: https://doi.org/10.1093/nar/gku1106.

**Questions:**

No questions.

---

> ### Author Response · Authors · 2024-11-26
> **Author Response (1/2)**
>
> We appreciate your comprehensive and constructive review. Our point-to-point responses to your comments are given below.
>
> ---
>
> **W1**. The method lacks novelty, as the clustering of canonical CDR loops has been previously established and used for antibody sequence design in frameworks such as Rosetta.
>
> **RW1**. Rosetta is a search-grafting energy-based technique used for local modifications of antibodies, but it relies heavily on human expertise and exhibits a constrained success rate. In contrast, IgSeek automates this process by leveraging extensive CDR templates, and rapidly identifying designable and scaffolding positions within CDRs. By accurately embedding CDR conformations into equal-length digital representations, IgSeek can seamlessly integrate its embeddings into other antibody generative models. We believe this capability represents the most significant innovation of our work.
>
> ---
>
> **W2**. The comparison with FoldSeek in Figure-2 is problematic; FoldSeek is designed for conserved regions of general proteins and is not suited for flexible loops such as antibody CDRs. Despite that, it’s surprising to me that FoldSeek outperformed IgSeek in CDR-H3.
>
> **RW2**. We appreciate your valuable feedback. It is important to clarify that utilizing FoldSeek to identify similar CDR conformations is a reasonable approach. Its proficiency in identifying similar CDR conformations is attributed to its utilization of dihedral angles between consecutive residues and its incorporation of 3Di alphabets to capture tertiary interactions among residues and their spatially closest neighbors. Essentially, FoldSeek can be characterized as a local alignment framework. On the other hand, IgSeek operates as a global alignment framework, generating fixed-length embeddings for distinct CDRs.
>
> Furthermore, previous research [1] has revealed that despite the extensive sequence diversity among antibodies, there exists a limited set of canonical structures within 5 out of 6 CDRs, and that certain CDR conformations are scaffolded by a few highly conserved residues, as mentioned in the Introduction Section of our original manuscript. This potentially contributes to the efficacy of FoldSeek in exploring hyper-variable antibody CDRs.
>
> Yet, it is important to highlight that our experiments have demonstrated that IgSeek outperformed FoldSeek in all other CDRs except for CDR-H3, while simultaneously achieving a notable 2.6x enhancement in terms of structure retrieval speed over other inverse folding methods.
>
> ---
>
> **W3**. In Figure-3, the comparisons with Antifold and AbMPNN for amino acid recovery tasks are unfair, as these models are not specifically trained for CDR loops alone. As acknowledged by the authors, this biases the results in favour of IgSeek, potentially exaggerating its performance.
>
> **RW3**. We acknowledge that AntiFold and AbMPNN are not developed within the same context of antibody design as our study. Inspired by Bennett et al. [2], our approach focuses on designing CDR loops using a unified antibody framework, which emphasizes good developability and aligns better with pharmaceutical practices than designing CDRs alongside the framework. Traditional methods often infer CDR sequences based on framework sequence information, which is less complex due to the interdependence between CDR1/CDR2 and the framework regions encoded by the same germline V gene. However, this can lead to data leakage and is less practical for pharmaceutical applications, where only antigen information is provided.
>
> To clarify the problem settings addressed in our work, we have included a new Table 1 in our revised manuscript, presenting a comparative analysis of various antibody design task configurations. Although we have attempted to identify published models trained under comparable conditions to IgSeek, we found that only the work from Baker’s group [2] aligns with this setting, and unfortunately, no source code has been publicly released. We welcome suggestions from reviewers regarding models that would facilitate a more informative comparison.
>
> ---
>
> **W4**. Figure-3 indicates that sorting by RMSD is more effective than using embeddings from IgSeek. Showing results based solely on RMSD (Kabsch) would provide a clearer performance baseline.
>
> **RW4**. We agree with the reviewer and will add the baseline performance using the Kabsch algorithm. While the Kabsch algorithm can compute ground truth RMSD values, its inference speed is **over two orders of magnitude** slower than IgSeek, posing challenges for its practical application in large-scale CDR database searches compared to IgSeek. In our experiment, we did not deploy the Kabsch algorithm to search the entire database. Instead, we validated the RMSD of the top-ranked CDRs identified by IgSeek until we identified the top 10 CDRs with RMSD < 1Å. This approach significantly reduced retrieval time while still leveraging the exact matching capabilities of the Kabsch algorithm.

---

> ### Author Response · Authors · 2024-11-26
> **Author Response (2/2)**
>
> ---
>
> **W5**. Figure-4 omits the inference time of the IgSeek+Kabsch method, although the authors use this method to argue that their approach outperforms others in Figure-3a/b. The added time from Kabsch alignment likely impacts overall inference speed, and a consistent comparison across methods is important.
>
> **RW5**. Thank you for your suggestion. We have included the inference time of the IgSeek+Kabsch in Figure 4 in our revised manuscript.
>
> ---
>
> **W6**. In the CDR sequence recovery tasks, the authors sample two sequences per query and select the one that best aligns with the ground truth, which is biased. In practical applications, such as library design for real experiments, the ground truth is unknown.
>
> **RW6**. Thank you for your comments. We want to clarify that existing protein and antibody inverse folding methodologies such as ProteinMPNN and AntiFold typically generate at least two samples for evaluation. In our work, we follow the settings of ProteinMPNN and present the best results of all other methods for evaluation.
>
> ---
>
> **W7**. IgSeek’s effectiveness heavily relies on the quality and diversity of the antibody structure database, which may restrict its applicability. This dependency also limits its ability to generate novel, diverse sequences, reducing its utility for de novo design.
>
> **RW7**. Thank you for your insightful feedback. One potential solution is to augment the database with predicted structures of a large-scale, diverse set of antibody sequences. Given the time constraints, this task will be deferred to our future work.
>
> ---
>
> **W8**. The paper does not specify how CDR regions are defined (what numbering scheme is used?)
>
> **RW8**. We apologize for the confusion regarding the numbering scheme. We used the IMGT numbering scheme:
> https://www.imgt.org/IMGTScientificChart/Numbering/IMGTIGVLsuperfamily.html
>
> ---
>
> **W9** Generated sequences for a given CDR loop are constrained to a fixed length, limiting flexibility for generating variable-length sequences, although similar limitations apply to existing methods.
>
> **RW9**. Thank you for your valuable feedback. We acknowledge that the constraint of generating sequences for a fixed-length CDR loop is a limitation shared by existing methods. To address this, we propose implementing our retrieval-based methodology on a significantly larger CDR database, for example, by using antibody structure prediction models to predict the sequences in the Observed Antibody Space (OAS). By leveraging this enriched dataset, we can effectively mitigate the identified limitation, enabling more flexible and robust solutions for sampling CDR loops of different lengths and enhancing the overall diversity and applicability of our approach.
>
> ---
>
> References:
>
> [1] C. Chothia et al. “Conformations of immunoglobulin hypervariable regions.” Nature, 342(6252):877–883, 1989.
>
> [2] N. R. Bennett et al. "Atomically accurate de novo design of single-domain antibodies." bioRxiv (2024).
>
> ---
>
> We sincerely appreciate your time, and we are glad to answer any additional questions you may have.

---

### Official Review · Reviewer_MRmb · 2024-11-09

**Soundness:** 2
**Presentation:** 3
**Contribution:** 2
**Rating:** 3
**Confidence:** 4

**Summary:**

This paper introduces IgSeek, a novel structure-retrieval framework that infers CDR sequences by retrieving similar structures from a natural antibody database. Specifically, IgSeek employs a simple yet effective multi-channel equivariant graph neural network
to generate high-quality geometric representations of CDR backbone structures. Then, it aligns sequences of structurally similar CDRs and utilizes structurally conserved sequence motifs to enhance inference accuracy. The authors claim that IgSeek is highly efficient in structural
retrieval and outperforms state-of-the-art approaches in sequence recovery for both antibodies and T-Cell Receptors.

**Strengths:**

* This paper has fast inference speed compared to existing inverse folding models like ESM-IF.

**Weaknesses:**

* The train/val/test split is based on time (year) and there can be identical or highly similar sequences in the training and test set. Therefore, there can be potential data leakage. Previous work like RefineGNN or DiffAb adopt sequence-similarity split to avoid this problem.
* Retrieving CDR sequences from known antibodies can lead to low novelty / non-specific binding of designed sequences. For example, if the task is to design CDRs that bind antigen A, IgSeek may retrieves a CDR from a known antibody that binds protein B. This CDR is unlikely to be a binder because it is designed specifically for protein B. And even if it does, it shows that this CDR is non-specific and this kind of non-specific binder is unfavorable. Therefore, casting protein design as a retrieval task has limitations.

**Questions:**

* Can you try construct a new train/val/test split based on CDR sequence similarity?

---

> ### Author Response · Authors · 2024-11-26
> **Author Response**
>
> We appreciate your comprehensive and constructive review. Our point-to-point responses to your comments are given below.
>
> ---
>
> **W1**. The train/val/test split is based on time (year) and there can be identical or highly similar sequences in the training and test set. Therefore, there can be potential data leakage. Previous work like RefineGNN or DiffAb adopt sequence-similarity split to avoid this problem.
>
> **RW1**. Thank you for your valuable feedback. In response, we have added a new figure to illustrate the sequence similarity comparisons between our training and test datasets. Please refer to Figure 7 in Appendix B for detailed statistics. As shown, the average sequence similarity across each CDR region between the training and test data ranges from 0.3 to 0.5. Additionally, we have taken care to remove any duplicated sequences from our dataset. This thorough data curation process has effectively minimized the risk of data leakage during model training.
>
> **W2**. Retrieving CDR sequences from known antibodies can lead to low novelty / non-specific binding of designed sequences. For example, if the task is to design CDRs that bind antigen A, IgSeek may retrieves a CDR from a known antibody that binds protein B. This CDR is unlikely to be a binder because it is designed specifically for protein B. And even if it does, it shows that this CDR is non-specific and this kind of non-specific binder is unfavorable. Therefore, casting protein design as a retrieval task has limitations.
>
> **RW2**. Thank you for your suggestion. We would like to take this opportunity to clarify the original motivation behind our research design. Many previous antibody design methods typically involve providing the sequence and structure of the antibody framework regions and training models to infer the sequences and structures of the CDRs. This approach inherently has a data leakage issue: since the sequences of CDR1 and CDR2, along with framework regions 1, 2, and 3, are encoded by the same germline V gene with minor somatic hypermutations, there is a strong interdependence between the sequences of CDR1/CDR2 and the framework regions. Moreover, this setup does not align well with pharmaceutical practices, where only the antigen information is given, and it is nontrivial for the generative model to determine which antibody framework to choose.
>
> In contrast, our method is inspired by a recent study by Bennett et al. [1], which designs different CDRs on the same framework to bind different antigen proteins. This framework has good developability, which can save a lot of subsequent optimization work and is more in line with pharmaceutical practice. This study uses a diffusion-based method [2] to generate the CDR backbone conformation, followed by sequence inference for this CDR conformation, known as inverse folding. Essentially, our work aims to enhance the accuracy of sequence inference in inverse folding through structure retrieval. As we discussed in our paper, the CDR conformation of antibodies is highly conserved and determined by specific amino acids. Therefore, the CDR conformation information can accurately determine the amino acid composition at specific positions that govern the shape of the CDRs.
>
> We are pleased to observe in our study that, even without antigen epitope information, our method can achieve an average AAR of around 0.5 based solely on the CDR backbone conformation. We hypothesize that the incorrect amino acids inferred by our model are due to the absence of the antigen epitope. These incorrect amino acids can be substituted with chemically complementary ones by experienced antibody engineers, or by incorporating antigen epitope information into the model for chemical complementarity reasoning in the future.
>
> Additionally, to overcome the limitations of the antibody CDR conformation database size, our method, due to its rapid inference time, can be extended to large-scale antibody CDR structure libraries predicted by AI structure prediction models like AlphaFold2 [3]. This allows us to retrieve diverse CDR sequences of different lengths to accommodate various antigen surfaces. This will be a significant research direction for our team in the future.
>
> ---
>
> **Q1**: Can you try construct a new train/val/test split based on CDR sequence similarity?
>
> **RQ1**: Please refer to **RW1**.
>
> ---
>
> Reference:
>
> [1] N. R. Bennett, et al. "Atomically accurate de novo design of single-domain antibodies." 2024, bioRxiv.
>
> [2] J. L. Watson, et al. "De novo design of protein structure and function with RFdiffusion." Nature 620(7976): 1089-1100, 2023.
>
> [3] J. Jumper, et al. "Highly accurate protein structure prediction with AlphaFold." Nature 596(7873): 583-589, 2021.
>
> ---
>
> We sincerely appreciate your time, and we are glad to answer any additional questions you may have.

---

### Meta-Review · Area_Chair_GjPh · 2024-12-20

**Metareview:**

The paper considers the design of functional CDR sequences by employing a retrieval based approach departing from inverse folding approaches that are often prone to hallucinations.

The paper tackle a relevant problem and overall is well written. However the novelty is somewhat limited as pointed out by multiple reviewers.

Also there are several issues which modulate the comparison against inverse-folding approaches. Indeed, key desiderata in sequence design include novelty/diversity and these are strongly gated by the database used for retrieval.  Augmenting the database with predicted structures as suggested by the authors is questionable as the quality might be problematic and one might face hallucination yet again.  Also only reporting AAR is insufficient in verifying that indeed hallucination has been reduced and that the generated sequences of good quality and faithful to the desired fold. The authors mentioned they would report other metrics and we strongly urge them to do so.

Important comparison are still pending (e.g. results based on RMSD (Kabsch) as mentioned by Reviewer L9vp).

Regarding the fixed-length CDR design, it is indeed a limitation shared with the comparison approaches considered. It would be interesting to include comparison with varying-length CDR design approaches.

**Additional Comments On Reviewer Discussion:**

The reviewers raised several important points pertaining to the validity of the experimental evaluation, limitations of the proposed approach, overemphasis of performance w.r.t. AAR, etc.  The authors have provided valuable feedback, e.g. to justify the validity of their data splitting, additional table comparing various approaches. However the authors continue to overemphasize performance in terms of AAR  (both in the paper and in their feedback e.g. in their response to Reviewer MRmb), have not provided satisfactory responses regarding the limitations of their approach to generate diverse and novel sequences (in response to reviewers MRmb and  tHJL).

Going forward we strongly encourage the authors to avoid referring the reviewers to the responses they provided to other reviewers, as it hinders a smooth and prompt discussion.

---

### Decision · Program_Chairs · 2025-01-22

Reject